# World Models via Policy-Guided Trajectory Diffusion

**Marc Rigter, Jun Yamada, and Ingmar Posner**

**Reviewed on OpenReview:** `https://openreview.net/forum?id=9CcgOOLhKG`

## Abstract

World models are a powerful tool for developing intelligent agents. By predicting the outcome of a sequence of actions, world models enable policies to be optimised via on-policy reinforcement learning (RL) using synthetic data, i.e. in "in imagination". Existing world models are autoregressive in that they interleave predicting the next state with sampling the next action from the policy. Prediction error inevitably compounds as the trajectory length grows. In this work, we propose a novel world modelling approach that is *not autoregressive* and generates entire on-policy trajectories in a single pass through a diffusion model. Our approach, Policy-Guided Trajectory Diffusion (PolyGRAD), leverages a denoising model in addition to the gradient of the action distribution of the policy to diffuse a trajectory of initially random states and actions into an on-policy synthetic trajectory. We analyse the connections between PolyGRAD, score-based generative models, and classifier-guided diffusion models. Our results demonstrate that PolyGRAD outperforms state-of-the-art baselines in terms of trajectory prediction error for short trajectories, with the exception of autoregressive diffusion. For short trajectories, PolyGRAD obtains similar errors to autoregressive diffusion, but with lower computational requirements. For long trajectories, PolyGRAD obtains comparable performance to baselines. Our experiments demonstrate that PolyGRAD enables performant policies to be trained via on-policy RL in imagination for MuJoCo continuous control domains. Thus, PolyGRAD introduces a new paradigm for accurate on-policy world modelling without autoregressive sampling.

## 1 Introduction

Model-based reinforcement learning (RL) trains a predictive model of the environment dynamics, conditioned upon actions. Such models are often referred to as *world models*. Once a world model has been learnt, imagined (i.e. synthetic) data generated from the world model can be utilised for planning (Schrittwieser et al., 2020) or on-policy RL (Hafner et al., 2021). By distilling the knowledge obtained about the environment into the world model, this approach facilitates improved sample efficiency relative to model-free approaches (Micheli et al., 2023) and zero-shot transfer to new tasks (Sekar et al., 2020).

A core challenge of this approach is learning a sufficiently accurate world model: If the world model is inaccurate, the imagined data will not be representative of the real environment, and the actions cannot be optimised effectively (Janner et al., 2019). Previous approaches have mitigated modelling errors by (a) using ensembles of models (Chua et al., 2018); (b) predicting the dynamics in latent space with a recurrent model (Ha & Schmidhuber, 2018; Hafner et al., 2021); or (c) using powerful generative models such as transformers (Micheli et al., 2023; Robine et al., 2023) or diffusion models (Anonymous, 2023) to generate accurate predictions. Common to all of these existing approaches is that they learn a *single-step* transition model, which conditions upon an action sampled from the current policy and predicts the next state. The model is unrolled autoregressively to generate imagined on-policy trajectories one step at a time. Modelling error accumulates at each step, meaning that the error in the trajectory inevitably grows as the trajectory length increases (Lambert et al., 2022). In this work, we propose an approach to world modelling that is *not autoregressive*, and generates entire on-policy trajectories in a single pass of diffusion.

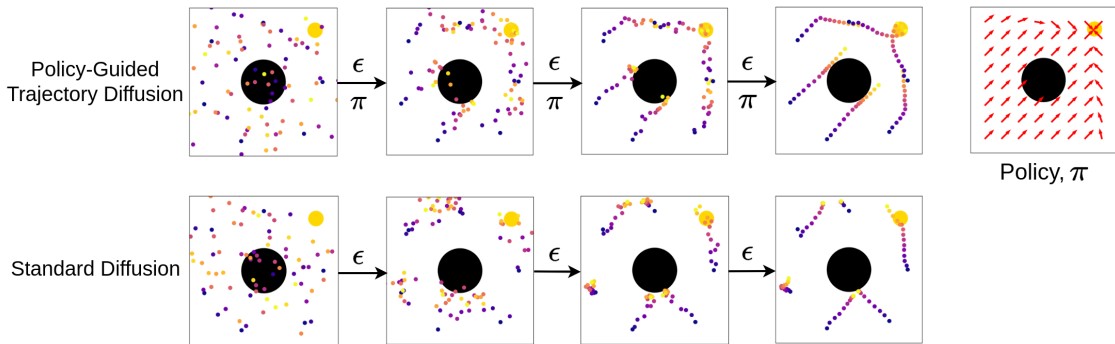

Figure 1: Top: Illustration of *Policy-Guided Trajectory Diffusion* (PolyGRAD). PolyGRAD starts with a trajectory of random states and actions and diffuses it into an on-policy trajectory using a learnt denoising model, $\epsilon$, and the policy, $\pi$. Bottom: Training a standard diffusion model on trajectories can be used to generate synthetic trajectories, but these are not on-policy.

The core challenge for creating a world model that is not autoregressive is ensuring that the trajectories are generated using actions sampled according to the current policy. In other words: *How do we sample actions from the policy at future states if these states have not yet been predicted?* This prohibits the standard approach of directly sampling actions from the policy. Instead, *Policy-Guided Trajectory Diffusion* (PolyGRAD) starts with a trajectory of random states and actions and gradually diffuses it into an on-policy synthetic trajectory (Figure 1). To accomplish this, PolyGRAD utilises a learnt denoising model, in addition to the gradient of the action distribution of the current policy. The denoising model is responsible for generating accurate predictions of the environment dynamics, while the policy is used to guide the diffusion process to on-policy trajectories. We analyse how our work can be viewed either as using a score-based generative model to generate on-policy actions, or as an instance of classifier-guided diffusion.

The core contributions of this work are: (a) proposing PolyGRAD, the first approach to world modelling that enables on-policy trajectory generation without autoregressive sampling, and (b) analysing the connection between PolyGRAD, score-based generative models, and classifier-guided diffusion models. Our results demonstrate that PolyGRAD outperforms state-of-the-art baselines in terms of trajectory prediction error for short trajectories, with the exception of autoregressive diffusion. For short trajectories, PolyGRAD obtains similar errors to autoregressive diffusion, but with lower computational requirements. For long trajectories, PolyGRAD obtains comparable performance to baselines. We show that PolyGRAD can be used to optimise performant policies via on-policy RL using only imagined data for MuJoCo continuous control domains. Thus, PolyGRAD introduces a new paradigm for developing accurate world models without autoregressive sampling.

## 2 Related Work

**World Models**  Model-based RL methods (Sutton, 1991) learn a model that predicts the transition dynamics and rewards in the environment, conditioned on previous observations and the current action. Such a model is commonly referred to as a "world model" (Ha & Schmidhuber, 2018; Kaiser et al., 2020). Earlier works on model-based RL used MLP ensembles (Chua et al., 2018; Yu et al., 2020) or Gaussian processes (Deisenroth & Rasmussen, 2011) to model the environment dynamics. More recently, the predominant approach has been to use a VAE (Kingma & Welling, 2013) to map observations to a compact latent space, and predict dynamics in the latent space using a recurrent model (Ha & Schmidhuber, 2018; Hafner et al., 2019; Doerr et al., 2018; Hafner et al., 2021). Such models are able to learn from high-dimensional observations in partially-observable environments. Transformers (Vaswani et al., 2017) have been used as more powerful sequence models for generating accurate dynamics predictions (Micheli et al., 2023; Robine et al., 2023; Schubert et al., 2023; Chen et al., 2022). Likewise, concurrently to our work, diffusion models have also been utilised to generate accurate predictions of individual state transitions in a world model (Anonymous, 2023). Once a world model has been learnt, it can be utilised for planning (Schrittwieser et al., 2020; Ye et al., 2021) or policy optimisation (Janner et al., 2019). Crucially, by conditioning on actions sampled from the current policy,

world models enable the generation of unlimited synthetic on-policy data, thus enabling on-policy RL "in imagination" (Hafner et al., 2019). The generation of on-policy synthetic trajectories also makes it possible to visualise the behaviour of the policy before deploying it (Wu et al., 2023). Learning policies entirely from synthetic data has the potential to improve sample efficiency (Micheli et al., 2023) and to facilitate transfer to new tasks without any further environment interaction (Sekar et al., 2020; Rigter et al., 2023).

Common to all of the aforementioned approaches is that they learn a *single-step* transition model, and use this model autoregressively to generate imagined trajectories of states and actions, one step at a time. In autoregressive models, error accumulates as the length of the prediction grows (Janner et al., 2019; Lambert et al., 2022) and therefore the length of the imagined trajectories is typically limited to a small number of steps (Janner et al., 2019; Hafner et al., 2021). Notably, Lambert et al. (2021) propose to learn a predictive model of environment dynamics that is not autoregressive. Instead, the model conditions upon the parameters of a simple controller, such as a PID or LQR controller, and is trained to predict the future state at an arbitrary point in time. Lambert et al. (2021) show that this approach achieves lower error at long-horizon predictions than simple autoregressive baselines. However, a limitation of Lambert et al. (2021) is that the controller parameters used to collect each transition in the dataset must be used as a training label. Therefore, this approach is not readily applicable to deep RL, where the number of policy parameters can be very large (i.e. millions). Like Lambert et al. (2021) we investigate an approach to trajectory generation that is not autoregressive. However, we aim to sample trajectories under a neural network policy defined by a large number of parameters. To enable this, we propose a novel world model that generates *entire on-policy trajectories in a single pass of diffusion.* To our knowledge, this is the first work to propose a method for non-autoregressive on-policy world modelling that is compatible with complex policies.

**Diffusion Models for Decision-making** Diffusion models are a powerful class of generative model that formulates data generation as an iterative denoising procedure (Sohl-Dickstein et al., 2015; Ho et al., 2020). Learning the denoising model is equivalent to learning the gradients of the data distribution, and therefore diffusion models are an instance of score-based generative models (Song et al., 2021). One advantage of diffusion models is that the iterative sampling procedure enables flexible conditioning, which is commonly utilised to generate bespoke high-quality synthetic images (Dhariwal & Nichol, 2021; Ho & Salimans, 2021).

In the context of sequential decision-making, the use of diffusion models was first introduced by Janner et al. (2022). The authors proposed to train a diffusion model to generate trajectories of states and actions, and use classifier-guidance to guide the trajectories towards goal states or high rewards thus enabling the diffusion model to be used as planner. This approach was subsequently extended to classifier-free guidance on state-only trajectories, with an inverse kinematics model to infer the actions (Ajay et al., 2023). Guided diffusion models have also proven to be highly effective in the multi-task setting (He et al., 2023). In each of these works, the output of the diffusion model is used to determine the action to be taken at each time step, making action selection very slow. Furthermore, unlike world models, trajectory generation cannot be conditioned on a specific policy or actions.

In another line of work, diffusion models have been used extensively as policies capable of accurately representing multi-modal action distributions. Such policies are particularly useful for offline RL (Hansen-Estruch et al., 2023; Wang et al., 2023) and imitation learning (Pearce et al., 2023; Chi et al., 2023), where it is important for the policy to remain near the action distribution in the dataset. In our work, we learn a separate feedforward policy that can be queried quickly during real-world rollouts, and use the diffusion model to generate on-policy synthetic data for policy training.

There are several previous works that use diffusion models to generate synthetic data for RL (Zhu et al., 2023). SynthER (Lu et al., 2023) trains a diffusion model to generate individual steps of off-policy synthetic data. This data is used to augment the replay buffer of an off-policy RL algorithm to improve sample efficiency. Ding et al. (2024) build on SynthER by generating off-policy trajectories rather than individual transitions. Similarly, MTDIFF-S (He et al., 2023) generates off-policy trajectories to augment offline RL datasets, but in the multi-task setting. Concurrent works on world modelling with diffusion (Anonymous, 2023; Zhang et al., 2023; Yang et al., 2023) train a diffusion model to generate a single step of transition data, conditioned on the current action and previous observations. Like other existing world models, this approach enables the generation of synthetic trajectories by sampling actions from the policy and autoregressively querying the

model one step at a time. In contrast, our approach generates entire on-policy trajectories in a single pass of diffusion. Also concurrently, and most related to our work, Jackson et al. (2023) guide diffusion with a policy to increase the likelihood of generated trajectories under the policy, in a similar manner to PolyGRAD. Jackson et al. (2023) use this approach to generate synthetic datasets for offline, off-policy RL. In contrast, PolyGRAD generates on-policy trajectories for online, on-policy RL in imagination. Unlike Jackson et al. (2023), we also analyse the connection between PolyGRAD, classifier-guidance, and score-based generative models. Furthermore, we show that the scale of the action guidance can be tuned automatically online to approximately generate the correct on-policy action distribution.

## 3 Preliminaries

Throughout this paper, we use subscript indices to refer to steps in a diffusion process and superscript indices to refer to time steps in a trajectory through the environment. Thus, $\mathbf{x}_i$ refers to a data sample at the $i^{\text{th}}$ step in a diffusion process, while $a^t$ refers to an action at the $t^{\text{th}}$ step in a trajectory. If a symbol has no subscript, it refers to a data sample that has had no noise added (i.e. $\mathbf{x} = \mathbf{x}_0$). Bold symbols refer to matrices.

**Denoising Diffusion Probabilistic Models**  Diffusion models (Ho et al., 2020; Sohl-Dickstein et al., 2015) are a class of generative models. Consider a sequence of positive noise scales, $0 < \beta_1, \beta_2, \ldots, \beta_N < 1$. In the forward process, for each training data point $\mathbf{x}_0 \sim p_{\text{data}}(\mathbf{x})$, a Markov chain $\mathbf{x}_0, \mathbf{x}_1, \ldots, \mathbf{x}_N$ is constructed such that $p(\mathbf{x}_i \mid \mathbf{x}_{i-1}) = \mathcal{N}(\mathbf{x}_i; \sqrt{1 - \beta_i}\mathbf{x}_{i-1}, \beta_i \mathbf{I})$. Therefore, $p_{\alpha_i}(\mathbf{x}_i \mid \mathbf{x}_0) = \mathcal{N}(\mathbf{x}_i; \sqrt{\alpha_i}\mathbf{x}_0, (1 - \alpha_i)\mathbf{I})$, where $\alpha_i := \Pi_{j=1}^i (1 - \beta_j)$. We denote the perturbed data distribution as $p_{\alpha_i}(\mathbf{x}_i) := \int p_{\text{data}}(\mathbf{x})p_{\alpha_i}(\mathbf{x}_i \mid \mathbf{x})\mathrm{d}\mathbf{x}$. The noise scales are chosen such that $\mathbf{x}_N$ is distributed according to $\mathcal{N}(\mathbf{0}, \mathbf{I})$. Define $s(\mathbf{x}_i, i)$ to be the score function of the perturbed data distribution: $s(\mathbf{x}_i, i) := \nabla_{\mathbf{x}_i} \log p_{\alpha_i}(\mathbf{x}_i)$, for all $i$. Samples can be generated from a diffusion model by starting from $\mathbf{x}_N \sim \mathcal{N}(\mathbf{0}, \mathbf{I})$ and following the recursion:

$$\mathbf{x}_{i-1} = \frac{1}{\sqrt{1 - \beta_i}}(\mathbf{x}_i + \beta_i s_\theta(\mathbf{x}_i, i)) + \sqrt{\beta_i}\mathbf{z}, \tag{1}$$

where $s_\theta$ is a learnt approximation to the true score function $s$, and $\mathbf{z}$ is a sample from the standard normal distribution. If we reparameterize the sampling of the noisy data points according to: $\mathbf{x}_i = \sqrt{\alpha_i}\mathbf{x}_0 + \sqrt{1 - \alpha_i}\boldsymbol{\epsilon}$, where $\boldsymbol{\epsilon} \sim \mathcal{N}(\mathbf{0}, \mathbf{I})$, we observe that

$$\nabla_{\mathbf{x}_i} \log p_{\alpha_i}(\mathbf{x}_i \mid \mathbf{x}_0) = -\frac{\boldsymbol{\epsilon}}{\sqrt{1 - \alpha_i}}. \tag{2}$$

Thus, estimating the score function is equivalent to estimating the noise added. Therefore, we can define the estimated score function in terms of a function $\epsilon_\theta$ that predicts the noise $\boldsymbol{\epsilon}$ used to generate each sample

$$s_\theta(\mathbf{x}_i, i) := -\frac{\epsilon_\theta(\mathbf{x}_i, i)}{\sqrt{1 - \alpha_i}}. \tag{3}$$

The noise prediction model $\epsilon_\theta$ is trained to optimise the objective

$$\theta^* = \arg\min_\theta \sum_{i=1}^N \mathbb{E}_{\mathbf{x}_0 \sim p_{\text{data}}(\mathbf{x})} \mathbb{E}_{\boldsymbol{\epsilon} \sim \mathcal{N}(\mathbf{0}, \mathbf{I})} \left[ ||\boldsymbol{\epsilon} - \epsilon_\theta(\sqrt{\alpha_i}\mathbf{x}_0 + \sqrt{1 - \alpha_i}\boldsymbol{\epsilon}, i)||^2 \right]. \tag{4}$$

**Sequential Decision-Making under Uncertainty**  We consider the setting of fully-observable Markov decision processes (MDPs). An MDP is defined by the tuple, $M = (S, A, T, R, \mu_0, \gamma)$. $S$ and $A$ are the state and action spaces, and $\mu_0$ is the initial state distribution. $T : S \times A \to \Delta(S)$ is the transition function, where $\Delta(X)$ denotes the set of possible distributions over $X$, and $\gamma$ is the discount factor. $R : S \times A \to \Delta(\mathbb{R})$ is the reward function. A policy, $\pi$, maps each state to a distribution over actions: $\pi : S \to \Delta(A)$. We will write $\boldsymbol{\tau} = s^0, a^0, r^0, \ldots, s^h, a^h, r^h$ to refer to a trajectory of states, actions, and rewards in an MDP with horizon $h$. $\boldsymbol{\tau}^a$ refers to the sequence of actions only, and $\boldsymbol{\tau}^{sr}$ refers to the sequence of states and rewards only. The standard objective for MDPs is to find the policy which maximises the total expected discounted reward.

**Model-Based Reinforcement Learning and World Models**  Model-based approaches to reinforcement learning (Sutton, 1991) (RL) utilise a predictive model of the environment dynamics, commonly referred to as a world model (Ha & Schmidhuber, 2018). During online rollouts, trajectories are collected from the

real environment and stored in data buffer $\mathcal{D}$. World models typically use $\mathcal{D}$ to learn a single-step transition model, $\widehat{T}$, as an approximation to the dynamics of the environment, $T$. If observations are high-dimensional, this dynamics model is usually learnt in a compressed latent representation of the state (Hafner et al., 2019). Additionally, a model of the reward function $\widehat{R}$ is also learnt from the data.

Once the world model has been trained, autoregressive sampling can be used to generate synthetic on-policy trajectories. To generate an imagined trajectory, $\widehat{\boldsymbol{\tau}}$, we first sample an initial state, $s^0$. We then sample an action from the policy conditioned on this state, $a^0 \sim \pi(\cdot|s^0)$. From the world model, we sample a reward $r_0 \sim \widehat{R}(\cdot|s^0, a^0)$ and a successor state, $s^1 \sim \widehat{T}(\cdot|s^0, a^0)$. This process is repeated step-by-step until a trajectory of the desired length has been generated: $\widehat{\boldsymbol{\tau}} = s^0, a^0, r^0 \ldots, s^h, a^h, r^h$. Because the world model is only an approximation to the environment, the error between imagined and real trajectories accumulates as the length of the trajectory grows. To mitigate this issue, small values of the horizon $h$ are typically used (Janner et al., 2019; Hafner et al., 2021). The imagined trajectories can then be utilised for online planning (Argenson & Dulac-Arnold, 2021; Schrittwieser et al., 2020) or on-policy RL (Hafner et al., 2019; 2021) to optimise decision-making. In this work, we will focus on the latter case of on-policy RL for policy optimisation.

## 4 World Models via Policy-Guided Trajectory Diffusion

In this work, we propose a new approach to world modelling: Policy-Guided tRAjectory Diffusion (PolyGRAD). A core novelty of PolyGRAD is that it enables the generation of entire on-policy trajectories in a single pass of diffusion, rather than autoregressively chaining a sequence of one-step predictions. The main challenge for creating a non-autoregressive world model is ensuring that the trajectories generated are sampled according to the current policy. To address this challenge, PolyGRAD utilises diffusion to gradually diffuse a trajectory of initially random states and actions into an on-policy synthetic trajectory.

PolyGRAD utilises two learned components: a denoising model, $\epsilon_\theta$, and a policy, $\pi_\phi$, defined by parameters $\theta$ and $\phi$ respectively. In this section, we first describe the denoising model, $\epsilon_\theta$. Second, we describe our main contribution, PolyGRAD, which uses the denoising model in conjunction with the policy to generate synthetic on-policy trajectories via diffusion. Finally, we provide an algorithm for imagined RL in PolyGRAD world models that automatically tunes the magnitude of the action updates in an outer loop.

### 4.1 Denoising Model Training (Algorithm 1)

Denoising model training in Algorithm 1 follows the same process as a standard diffusion model, with the exception that we add action-conditioning. To train the denoising model, we sample a sequence of states and rewards, $\boldsymbol{\tau}^{sr} = s^0, r^0 \ldots, s^h, r^h$, and a sequence of actions, $\boldsymbol{\tau}^a = a^0, \ldots, a^h$, from the data buffer, $\mathcal{D}$. The step in the diffusion process, $i$, is also sampled. Random noise $\sqrt{1 - \alpha_i}\epsilon$ is added to the state and reward sequence, where $\epsilon$ is sampled according to a standard normal distribution and $\alpha_i$ is defined by the diffusion noise schedule. The denoising model is trained to predict the noise added to the original state and reward sequence, conditioned upon the actions and the diffusion step. In our implementation, the denoising model is

---

**Algorithm 1** Denoising Model Training

1: **Require**: denoising model, $\epsilon_\theta$; data buffer, $\mathcal{D}$; number of diffusion steps, $N$

2: **while** training :
3: $\quad (\boldsymbol{\tau}^{sr}, \boldsymbol{\tau}^a) \sim \mathcal{D}$
4: $\quad i \sim \text{Uniform}(1, N)$
5: $\quad \epsilon \sim \mathcal{N}(\mathbf{0}, \mathbf{I})$
6: $\quad$ Take gradient descent step on

$$\nabla_\theta ||\epsilon - \epsilon_\theta(\sqrt{\alpha_i}\boldsymbol{\tau}^{sr} + \sqrt{1 - \alpha_i}\epsilon \mid i, \boldsymbol{\tau}^a)||^2$$

---

also trained to predict whether a state is terminal, but we omit this from our notation to avoid clutter.

### 4.2 Policy-Guided Trajectory Diffusion (Algorithm 2)

We now present PolyGRAD, our algorithm for generating imagined on-policy trajectories for world modelling (Algorithm 2). Each of the key steps of the algorithm is illustrated in Figure 2. PolyGRAD begins with a trajectory of random states, rewards, and actions, $\widehat{\boldsymbol{\tau}}_N = (\widehat{\boldsymbol{\tau}}_N^{sr}, \widehat{\boldsymbol{\tau}}_N^a)$, sampled from a standard normal distribution (Lines 2 and 3). Note that the subscript $N$ refers to the step in the diffusion process. We also sample an initial state $s^0$ uniformly from $\mathcal{D}$. The trajectory is then iteratively refined over many diffusion steps. To condition the trajectory on the initial state from the dataset $s^0$, we perform inpainting (Lugmayr

---

**Algorithm 2** Policy-Guided Trajectory Diffusion (PolyGRAD)

---

1: **Require**: policy, $\pi_\phi$; denoising model, $\epsilon_\theta$; action update scale, $\delta$; data buffer $\mathcal{D}$
2: $\widehat{\boldsymbol{\tau}}_N^a \sim \mathcal{N}(\mathbf{0}, \mathbf{I})$
3: $\widehat{\boldsymbol{\tau}}_N^{sr} \sim \mathcal{N}(\mathbf{0}, \mathbf{I})$
4: $s^0 \sim \mathcal{D}$
5: **for** $i = N, N-1, \ldots, 1$ :
6:     set initial state in $\widehat{\boldsymbol{\tau}}_i^{sr}$ to $s^0$           ▷ Condition trajectory on initial state from dataset
7:     $\widehat{\boldsymbol{\epsilon}} \leftarrow \epsilon_\theta(\widehat{\boldsymbol{\tau}}_i^{sr} \mid i, \widehat{\boldsymbol{\tau}}_i^a)$           ▷ Predict noise
8:     **if** $i > 1$ :
9:         $\widehat{\boldsymbol{\tau}}_0^{sr} \leftarrow \frac{1}{\sqrt{\alpha_i}} \cdot \widehat{\boldsymbol{\tau}}_i^{sr} - \frac{\sqrt{1-\alpha_i}}{\sqrt{\alpha_i}} \cdot \widehat{\boldsymbol{\epsilon}}$       ▷ Predict fully-denoised state sequence
10:         $\widehat{\boldsymbol{\tau}}_{i-1}^a \leftarrow \widehat{\boldsymbol{\tau}}_i^a + \delta \cdot \nabla_{\widehat{\boldsymbol{\tau}}_i^a} \log \pi_\phi(\widehat{\boldsymbol{\tau}}_i^a \mid \widehat{\boldsymbol{\tau}}_0^s) + \sqrt{\beta_i}\mathbf{z}$   ▷ Update actions using policy score
11:     **else** :
12:         $\widehat{\boldsymbol{\tau}}_{i-1}^a \leftarrow \widehat{\boldsymbol{\tau}}_i^a$
13:     $\widehat{\boldsymbol{\tau}}_{i-1}^{sr} \leftarrow \frac{1}{\sqrt{1-\beta_i}}(\widehat{\boldsymbol{\tau}}_i^{sr} - \frac{\beta_i}{\sqrt{1-\alpha_i}}\widehat{\boldsymbol{\epsilon}}) + \sqrt{\beta_i}\mathbf{z}$   ▷ Update states and rewards using diffusion (Equation 1)
14: **return** $\widehat{\boldsymbol{\tau}} = (\widehat{\boldsymbol{\tau}}_0^{sr}, \widehat{\boldsymbol{\tau}}_0^a)$

---

et al., 2022) by replacing the initial state in the trajectory by $s^0$ at each step of the diffusion process (Line 6). Generating rollouts branched from states in the dataset is standard practice in model-based RL (Janner et al., 2019). At each diffusion step, the denoising model is used to predict the noise added to the state and reward sequence, conditioned on the current action sequence (Line 7). The predicted noise is denoted by $\widehat{\boldsymbol{\epsilon}}$.

In all steps except for the final diffusion step, the action sequence is updated in Lines 9 and 10. To perform the action update, we first use the predicted noise $\widehat{\boldsymbol{\epsilon}}$ to compute a prediction of the fully-denoised state sequence, $\widehat{\boldsymbol{\tau}}_0^s$. We compute this prediction because the policy is trained on data without noise added, so we do not wish to condition the policy on a noisy state sequence. In Line 10, we condition the policy on the predicted denoised state sequence. We update the action sequence in the direction of the score of the policy action distribution, $\nabla_a \log \pi_\phi(a|s)$, in Line 10. Note that this gradient differs from the standard gradient used in policy-gradient RL algorithms (which is $\nabla_\phi \log \pi_\phi(a|s)$), and can be easily computed for common policy parameterisations, such as Gaussian policies. The magnitude of the update to the actions is controlled by the action update scale, $\delta$, which as we shall see in Section 4.3, is tuned automatically online.

In Line 13 the state and reward sequence is updated using the noise prediction and a standard diffusion update (Equation 1). After completing all the diffusion steps, the final synthetic trajectory is returned in Line 14.

In summary, the action sequence is updated using the gradient of the policy action distribution $\nabla_a \log \pi_\phi(a|s)$ to increase the likelihood of the actions under the policy. The state and reward sequence is updated according to the learnt denoising model. We must ensure that the state and action sequences are consistent (i.e. the actions are sampled according to the policy conditioned on the predicted states, and the states are predicted according to the sampled actions). To ensure this consistency, the policy is conditioned on the current predicted denoised state sequence, and the denoising model is conditioned on the current action sequence. Provided that the action update size $\delta$ is chosen correctly, during diffusion the actions are iteratively updated until they are similar to samples drawn from the policy. The state-reward sequence is predicted conditioned upon those actions. Therefore, PolyGRAD is able to generate on-policy synthetic trajectories.

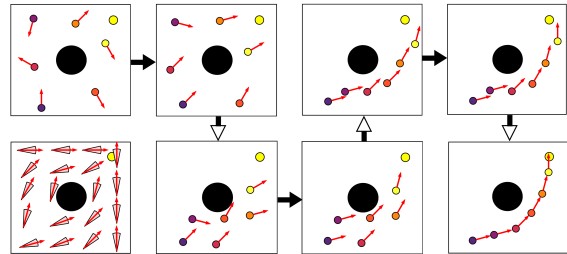

Figure 2: Step-by-step illustration of PolyGRAD trajectory generation (Algorithm 2). Bottom left: Illustration of policy action distribution throughout state space. Top left: Trajectory is initialised with random states and actions (Line 2 and 3). The initial state (dark purple) is sampled from the dataset (Line 4). Solid black arrows: Action sequence is updated according to score of policy conditioned on current state sequence (Line 10). Hollow black arrows: State sequence is updated conditioned on current actions (Line 13). Bottom right: Final trajectory is returned.

Now that we have described the mechanics of our algorithm, we provide theoretical motivation for PolyGRAD by comparing it to score-based generative models and classifier-guided sampling.

**Connection to Score-Based Generative Models** Langevin dynamics (Rossky et al., 1978; Neal, 2011) is a Markov chain Monte Carlo method that produces samples from a probability density $p(\mathbf{x})$ using only the score function $\nabla_{\mathbf{x}} \log p(\mathbf{x})$. Given a fixed step size $\beta > 0$, and an arbitrary initial value $\mathbf{x}_0$, the Langevin method recursively computes

$$\mathbf{x}_t = \mathbf{x}_{t-1} + \frac{\beta}{2} \nabla_{\mathbf{x}} \log p(\mathbf{x}_{t-1}) + \sqrt{\beta} \mathbf{z}, \quad \mathbf{z} \sim \mathcal{N}(0, I). \tag{5}$$

When $\beta \to 0$ and $t \to \infty$, Langevin dynamics converges to a sample from $p(\mathbf{x})$ (Song & Ermon, 2019; Neal, 2011). Score-based generative models attempt to learn an approximation to the score function, $s_\theta(\mathbf{x}) \approx \nabla_{\mathbf{x}} \log p(\mathbf{x})$, and then perform sampling according to Equation 5 using $s_\theta$. As noted in Section 3, diffusion models are in instance of score-based generative models, where the score of the noisy data distribution is learnt across a range of noise levels.

In Line 10 of Algorithm 2, we update the action sequence using the same update rule as the Langevin method (Equation 5), but with a few changes. Unlike a standard score-based generative model, where an approximation to the score function is learned, in PolyGRAD we directly use the score function of the policy action distribution, $\nabla_a \log \pi(a|s)$. Furthermore, the policy score function is conditioned upon the current predicted denoised state sequence, $\hat{\boldsymbol{\tau}}_0^s$ (which also changes throughout the diffusion process). The final modification is that we allow the update size to be a tuneable parameter, $\delta$. Because the action sequence is updated according Langevin dynamics, we might expect it should converge to a sample near the policy action distribution, provided that: a) the state sequence $\hat{\boldsymbol{\tau}}_0^s$ converges, b) the number of diffusion steps is large, and c) $\delta$ and $\beta$ are sized appropriately. A formal analysis of convergence is outside the scope of this work.

**Connection to Classifier-Guidance** In classifier-guided sampling (Dhariwal & Nichol, 2021), the aim is to sample from the conditional distribution $p(\mathbf{x} \mid y)$, where $y$ is a label. Recall that $p_{\alpha_i}$ denotes the noisy data distribution in the $i^{\text{th}}$ step of a diffusion model. To enable sampling from $p(\mathbf{x} \mid y)$ using diffusion (Equation 1), instead of approximating the score function $\nabla_{\mathbf{x}_i} \log p_{\alpha_i}(\mathbf{x})$, we would like to approximate the score function $\nabla_{\mathbf{x}_i} \log p_{\alpha_i}(\mathbf{x}_i \mid y)$. From Bayes' rule, we have that

$$\nabla_{\mathbf{x}_i} \log p_{\alpha_i}(\mathbf{x}_i \mid y) = \nabla_{\mathbf{x}_i} \log p_{\alpha_i}(\mathbf{x}_i) + \nabla_{\mathbf{x}_i} \log p(y \mid \mathbf{x}_i).$$

Therefore, if we have a classifier $p(y \mid \mathbf{x}_i)$, the gradient of the classifier with respect to $\mathbf{x}_i$ can be used to guide the diffusion process to sample from the conditional distribution $p(\mathbf{x} \mid y)$. Note that the classifier is evaluated on noisy samples, $\mathbf{x}_i$, so the classifier is typically also trained on noisy samples.

In our work, we are interested in sampling on-policy trajectories from the distribution $p(\boldsymbol{\tau} \mid \pi_\phi)$, where $\pi_\phi$ is the current policy. To directly apply classifier-guided sampling to this problem, we would need to learn a differentiable classifier $p(\pi_\phi \mid \boldsymbol{\tau}_i)$ so that we can evaluate $\nabla_{\boldsymbol{\tau}_i} \log p(\pi_\phi \mid \boldsymbol{\tau}_i)$. However, it is unclear how to train such a classifier. To overcome this issue, let us consider $\boldsymbol{\tau}$, a trajectory that has had no noise added. Applying Bayes' rule, we have that

$$p(\pi_\phi \mid \boldsymbol{\tau}) = \frac{p(\pi_\phi, \boldsymbol{\tau})}{p(\boldsymbol{\tau})} = \frac{p(\pi_\phi)\mu(s^0)\pi_\phi(a^0|s^0)R(r^0|s^0, a^0)T(s^1|s^0, a^0)\dots}{\mu(s^0)p(a^0|s^0)R(r^0|s^0, a^0)T(s^1|s^0, a^0)\dots} = \frac{p(\pi_\phi)\pi_\phi(a^0|s^0)\pi_\phi(a^1|s^1)\dots}{p(a^0|s^0)p(a^1|s^1)\dots}, \tag{6}$$

where $p(a|s)$ represents the probability of action $a$ being selected at state $s$ under any policy. Taking the gradient with respect to $\boldsymbol{\tau}$, we have

$$\nabla_{\boldsymbol{\tau}} \log p(\pi_\phi \mid \boldsymbol{\tau}) = \sum_{i=0}^{h} \nabla_{\boldsymbol{\tau}} \log \pi_\phi(a^i|s^i) - \sum_{i=0}^{h} \nabla_{\boldsymbol{\tau}} \log p(a^i|s^i). \tag{7}$$

If we assume that $p(a|s)$ is a uniform distribution over actions for all $s$, then the second term is zero. Therefore, under this assumption that by default all actions are equally likely, as well as the assumption that no noise

has been added to the trajectory, we arrive at the final classifier gradient:

$$\nabla_{\boldsymbol{\tau}} \log p(\pi_\phi \mid \boldsymbol{\tau}) = \sum_i \nabla_{\boldsymbol{\tau}} \log \pi_\phi(a^i | s^i). \tag{8}$$

Deriving the correct classifier-guidance gradient for the case where noise has been added to the trajectory is a subject for future work. To avoid conditioning the policy on noisy trajectories, in PolyGRAD we use the denoising model to predict the fully denoised state sequence, $\widehat{\boldsymbol{\tau}}_0^s$, and condition the policy on this sequence to compute the score function (Line 9 of Algorithm 2).

Thus, by considering classifier guidance we arrive at a similar update rule for the actions to that used by PolyGRAD. However, the classifier-guidance inspired update in Equation 8 indicates that both actions *and* states in the trajectory should be updated in the direction of the score of the policy distribution. Meanwhile, in Line 10 of the PolyGRAD algorithm (Algorithm 2), we update only the action sequence in this manner. The state sequence is updated using the denoising model only. In the experiments, we assess the performance of PolyGRAD when both the action and state sequences are updated according to Equation 8.

### 4.3   Imagined RL in PolyGRAD World Models (Algorithm 3)

Following previous works on world models for RL (Hafner et al., 2019; 2021), we optimise the policy by performing on-policy RL on the imagined trajectories generated by PolyGRAD. We assume that for each state, the policy $\pi_\phi$ outputs a Gaussian distribution over actions: $\pi_\phi(a|s) = \mathcal{N}(\mu_\phi(s), \sigma_\phi(s))$. This is the most commonly used policy parameterisation in deep RL (Schulman et al., 2017; Haarnoja et al., 2018). For the most part, Algorithm 3 follows a standard model-based RL training loop. Data is gathered from the real environment, and used to train the denoising model $\epsilon_\theta$. PolyGRAD is then used to generate a batch of

---

**Algorithm 3** Imagined RL in PolyGRAD World Model

1: **Require**: environment, $E$;
2: **Initialise**: policy, $\pi_\phi$; denoising model $\epsilon_\theta$; action update scale, $\delta$; empty data buffer, $\mathcal{D}$

3: **while** training :
4:    $\boldsymbol{\tau} \leftarrow E(\pi_\phi)$          ▷ Sample real trajectory
5:    $\mathcal{D}.\texttt{add}(\boldsymbol{\tau})$
6:    Train $\epsilon_\theta$ on $\mathcal{D}$          ▷ Algorithm 1
7:    Generate synthetic trajectories, $\{\widehat{\boldsymbol{\tau}}\}$ ▷ Algorithm 2
8:    Train $\pi_\phi$ on $\{\widehat{\boldsymbol{\tau}}\}$ via on-policy RL
9:    Update action update scale, $\delta$      ▷ Equation 10

---

on-policy synthetic trajectories. These trajectories are used to update the policy via on-policy RL. The key difference with respect to standard model-based RL is that Line 9 of Algorithm 3 updates $\delta$, which controls the magnitude of the action updates in PolyGRAD.

To perform the update for $\delta$, we consider the set of state-action pairs in the synthetic trajectories generated by PolyGRAD, $\{\widehat{\boldsymbol{\tau}}\}$. For each state-action pair, $(s_i, a_i)$, we standardise the action according to the mean and standard deviation of the policy action distribution at that state:

$$\bar{a}_i = \frac{a_i - \mu_\phi(s_i)}{\sigma_\phi(s_i)}. \tag{9}$$

We then compute $\sigma_{\bar{a}}$, the standard deviation of the set of standardised actions $\{\bar{a}\}$. If the actions are drawn correctly from the policy distribution, then the standardised actions should be distributed according to a standard normal distribution. Therefore, we update the action update scale to obtain a standard deviation of 1 in the standardised actions:

$$\delta \leftarrow \delta + \eta \cdot (\sigma_{\bar{a}} - 1), \tag{10}$$

where $\eta$ is a learning rate. The intuition for Equation 10 is as follows. If the policy guidance is too strong, all of the actions will be guided to be very near to the mean of the policy distribution. Therefore, the standardised actions will have a variance that is too low (i.e. below 1) and Equation 10 will reduce the action update scale. Likewise, if the guidance is too weak, the actions will remain spread out far from the policy mean. Thus, the standardised actions will have a variance that is too high and Equation 10 will increase the action update scale. As we shall show in the experiments, we found that tuning the action update sizing in this manner is sufficient to ensure the PolyGRAD generates a good approximation of the correct on-policy action distribution.

**Implementation Details** Here, we outline some key implementation details. A detailed description of our implementation is in Appendix A. After performing each update to the actions in Line 10 of Algorithm 2, we clip the actions to be within 3 standard deviations of the mean of the policy action distribution. We found that this helps to ensure that the action distribution can be reliably guided to the correct on-policy distribution. For imagined RL training in Algorithm 3, we used Advantage Actor Critic (A2C) (Mnih et al., 2016) with Generalised Advantage Estimation (GAE) (Schulman et al., 2016). We observed that RL training with PolyGRAD world models is unstable when there are large updates to the policy: if the policy is changed drastically, PolyGRAD may not consistently produce the correct action distribution for on-policy RL training, leading to policy collapse. Therefore, we decay the learning rate for the policy to maintain a constant update size. For the denoising network, we use either a residual MLP or a transformer. As described in Algorithm 1, the denoising network is trained using actions that have no noise added, despite the fact that it is evaluated on (initially) noisy actions in Algorithm 2. However, we found this obtained much better prediction errors and RL performance than training the denoising network on noisy actions.

## 5 Experiments

In our experiments, we seek to answer the following questions: (a) Does PolyGRAD produce the correct action distribution? (b) How accurate are the trajectories produced by PolyGRAD compared to autoregressive world models? (c) Can the synthetic data produced by PolyGRAD be used to train performant policies? (d) Which implementation details influence the performance of PolyGRAD? To answer these questions, we run experiments using the MuJoCo environments in OpenAI Gym (Brockman et al., 2016). The code for our experiments is available at github.com/marc-rigter/polygrad-world-models.

### 5.1 Does PolyGRAD produce the correct action distribution?

We wish to evaluate whether PolyGRAD generates synthetic trajectories with the correct on-policy action distribution. To evaluate the action distribution produced, we train an RL policy using synthetic data generated by PolyGRAD (Algorithm 3). We linearly decay the standard deviation of the Gaussian policy, $\sigma_\phi$, from 1 to 0.02 over the course of training. In Figure 3, we plot the distribution of the difference between the action $a$ and the mean of the action distribution, $\mu_\phi(s)$, over all state-action pairs $(s, a)$ in batches of data produced by PolyGRAD with $h = 50$ in Walker2d.

We observe that for $\sigma_\phi \geq 0.1$, the action distribution closely matches the target Gaussian distribution. This demonstrates that PolyGRAD produces the correct action distribution on aggregate provided that the policy entropy is not too low. However, we observe that as the policy standard deviation decreases below 0.1, the action distribution begins to deviate from the target distribution. Thus, if the policy has very low entropy, it is difficult for PolyGRAD to guide the initially randomly sampled actions to the target distribution. In the plot with $\sigma_\phi = 0.02$, the distribution has additional modes at $\pm 3$ standard deviations. This is an artifact of the action clipping described in the implementation details. Plots for the other MuJoCo environments are provided in Appendix B.3 of the supplementary material.

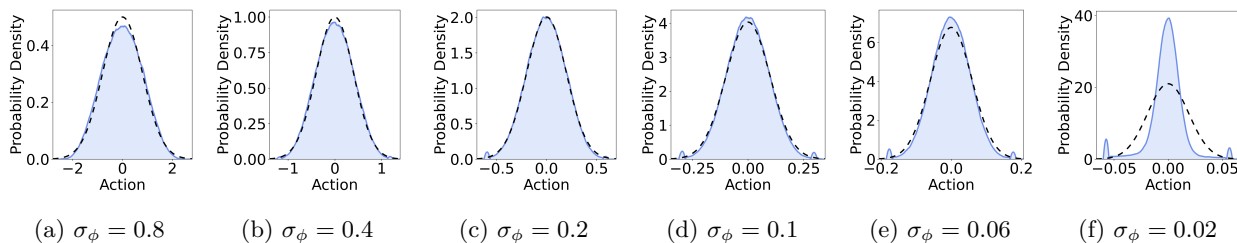

Figure 3: Plots of action distributions produced by PolyGRAD. Blue line illustrates the distribution of $a - \mu_\phi(s)$ for a batch of synthetic data. Each subplot is for a policy with a different entropy level that is constant throughout the state space. Dashed black line indicates the action distribution output by the policy. Data is generated by running Algorithm 3 in Walker2d with $h = 50$.

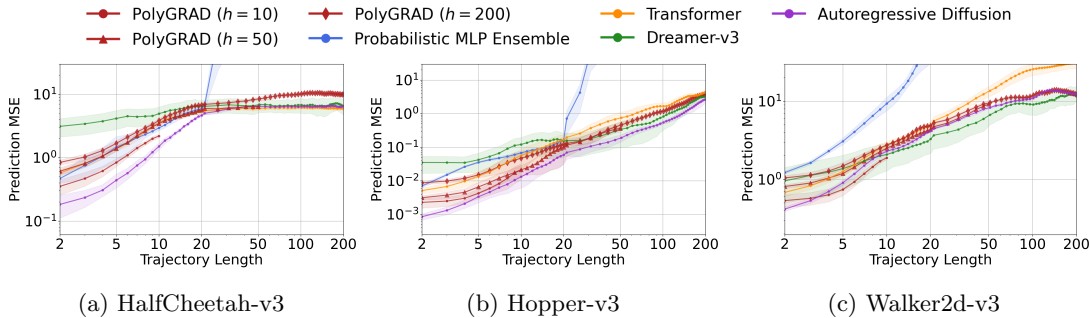

| (a) HalfCheetah-v3 | (b) Hopper-v3 | (c) Walker2d-v3 |

Figure 5: Plots of mean squared error (MSE) of predicted states vs ground truth states for each world model trained on the same dataset for each environment. Shaded regions indicate standard deviation over 5 seeds. For PolyGRAD, we use the transformer denoising network trained on trajectories of length $h = 10, 50$, or $200$.

## 5.2 How accurate are the trajectories produced by PolyGRAD compared to existing world models?

To assess the accuracy of the trajectories produced by PolyGRAD, we train several different world models using the same dataset of 1M transitions in each environment. We then generate synthetic rollouts using the same performant policy in each world model. We replay the same actions in the MuJoCo simulator, and evaluate the errors in the trajectory predictions. We compare the accuracy of the trajectories against the following baselines which all generate trajectories autoregressively:

- Probabilistic MLP Ensemble, an ensemble of MLPs outputting a Gaussian over the next state (Chua et al., 2018; Yu et al., 2020; Janner et al., 2019).
- Transformer, a transformer-based world model (Micheli et al., 2023).
- Dreamer-v3 (Hafner et al., 2023), a world model based on a sequential VAE.
- Autoregressive Diffusion, a diffusion model trained to generate one-step action-conditioned predictions, rolled out autoregressively (Anonymous, 2023; Zhang et al., 2023).

Detailed descriptions of the implementation of each baseline can be found in Appendix C.3, and further details on the experimental setup can be found in Appendix C.1.

Figure 5 shows the error for each world modelling method for varying trajectory lengths. We train PolyGRAD for trajectories of $h = 10, 50$, and $200$. PolyGRAD with $h = 10$ obtains the best trajectory errors for Walker-2d. For HalfCheetah and Hopper, PolyGRAD with $h = 10$ obtains the second-best errors to Autoregressive Diffusion. When PolyGRAD is trained on longer trajectories ($h = 50$ and $h = 200$) we observe that larger errors are obtained, and PolyGRAD performs less well relative to the baselines. This indicates that it is more challenging for the denoising model to accurately predict the denoised states for longer trajectories.

The errors for the Probabilistic MLP Ensemble increase quickly as the prediction horizon increases. This may be because this method outputs a Gaussian distribution over the next state, and repeatedly sampling from this Gaussian can quickly lead to out-of-distribution states. The Transformer baseline obtains comparable performance to PolyGRAD with $h = 200$, but PolyGRAD outperforms the transformer when trained and evaluated on shorter trajectories. Note that PolyGRAD uses the same transformer network architecture for the denoising model as the Transformer baseline. Therefore, this result demonstrates that there is an advantage to iteratively refining the trajectory via diffusion rather than autoregressively predicting each successor state with a transformer.

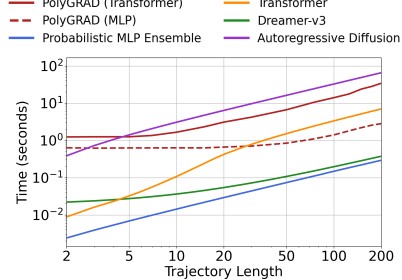

Figure 4: Computation times to produce a batch of 1000 trajectories on a V100 GPU in Walker2d.

Interestingly, while Dreamer-v3 performs well on Walker2d, it obtains the poorest prediction errors at short horizons for HalfCheetah and Hopper despite being a state-of-the-art world model that produces strong RL agents. This may be because Dreamer-v3 performs dynamics predictions and RL training in latent space, and the reconstruction error to the original state space forms only one part of the training objective. Finally,

Autoregressive Diffusion tends to be the most performant approach, especially on HalfCheetah. This indicates that for these MuJoCo environments, training a very accurate one-step prediction model leads to accurate trajectories even at longer horizons. However, the downside of Autoregressive Diffusion is that it is the slowest world modelling method (Figure 4) as it requires running diffusion at every step of the trajectory rollout, whereas PolyGRAD requires only a single pass of diffusion.

### 5.3 Can the synthetic data produced by PolyGRAD be used to train performant policies?

Following from previous works on model-based RL (Janner et al., 2019; Hafner et al., 2021; Yu et al., 2020), we use short imagined rollouts for model-based RL training ($h = 10$). For RL training, we use the residual MLP denoising network as it obtains similar error to the transformer denoising network at $h = 10$ (see Figure 10 in Appendix B.1) but it is faster for both training and inference (Figure 4).

To assess the performance of RL training in imagination for PolyGRAD (Algorithm 3), we compare the rewards obtained against model-free on-policy RL algorithms (PPO (Schulman et al., 2017), TRPO (Schulman et al., 2015), and A2C (Mnih et al., 2016)). We compare against on-policy model-free RL algorithms because we perform imagined on-policy training in the PolyGRAD world model. However, note that off-policy model-free RL algorithms obtain stronger performance for these environments (Haarnoja et al., 2018). To compare against a state-of-the-art world model-based agent, we also compare against Dreamer-v3 (Hafner et al., 2023). Reward curves for each algorithm are shown in Figure 6. Figure 7 provides 95% confidence intervals of the interquartile mean of the normalised final performance aggregated across all three environments (see Appendix C.2 for details). Figure 7 shows that PolyGRAD outperforms on-policy model-free algorithms. This is likely because PolyGRAD leverages all of the training data available to train the world model and generate large quantities of synthetic training data. On the other hand, on-policy model-free algorithms do not reuse training data. Confidence intervals in Appendix B.2 show that PolyGRAD significantly outperforms model-free algorithms in terms of mean performance in addition to interquartile mean.

PolyGRAD obtains worse sample-efficiency and final performance compared to Dreamer-v3, despite the fact that PolyGRAD obtains better prediction errors than Dreamer-v3 (Figure 5). This may be because Dreamer optimises policies by directly backpropagating through the dynamics model, while we train policies using policy-gradient RL. Alternatively, it may be because by optimising policies in latent space, Dreamer is able to produce more performant policies despite having less accurate reconstructions of the original state.

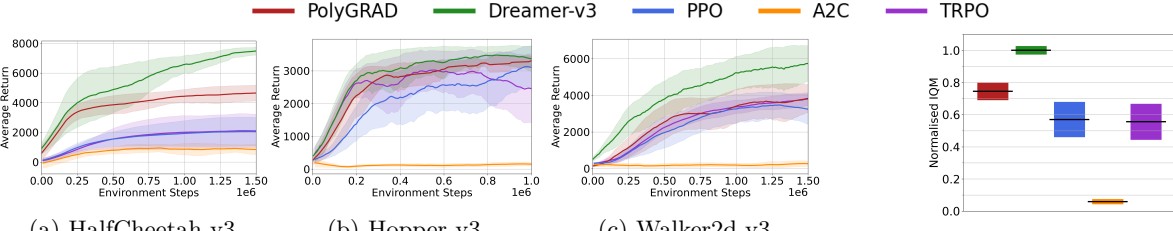

(a) HalfCheetah-v3    (b) Hopper-v3    (c) Walker2d-v3

Figure 6: Reward curves for RL training. Shaded regions indicate standard deviation over 5 seeds.

Figure 7: 95% CIs of interquartile mean of normalised final performance aggregated across envs.

### 5.4 Which implementation details influence the performance of PolyGRAD?

To assess which implementation details are important, we compare the performance of the base PolyGRAD algorithm against the following ablations and modifications:

- Random Actions: We sample random actions at the beginning of the diffusion process from a standard normal distribution, and they are not updated during diffusion.
- Policy Sampling: During each step of the diffusion process, we obtain new actions by directly sampling them from the policy conditioned upon the current predicted state sequence.
- No Clipping: The actions are not clipped during the diffusion process.

- Add State Update: In addition to updating the actions according to the score of the policy distribution, we also update the states to increase the likelihood of the actions according to Equation 8.
- Noisy State Conditioning: Instead of conditioning the policy on the denoised state prediction ($\widehat{\boldsymbol{\tau}}_0^a$ in Line 10 of Algorithm 2) we condition the policy on the noisy states ($\widehat{\boldsymbol{\tau}}_i^a$).

A detailed description of each of these modifications is provided in Appendix C.4. Figure 8 shows the reward curves for each variant of the algorithm, and Figure 9 shows 95% confidence intervals for the interquartile mean of the final normalised performance across all environments. We observe that Random Actions obtains the poorest performance. This is unsurprising as we use on-policy RL to optimise the policy, and therefore randomly sampled off-policy actions result in incorrect policy gradient updates. Policy Sampling also performs very poorly. This method also fails to generate on-policy trajectories due to the following reasoning. Imagine that we sample an entire sequence of actions, conditioned on the current state sequence. Then, conditioned on these new actions, a new prediction of the state sequence is generated at the next diffusion step. However, the policy has a different action distribution at the new state sequence. Thus, the final trajectory of states and actions produced is not on-policy. We also observed that Policy Sampling results in worse prediction errors. This is likely because a diffusion model, which gradually refines its predictions, is ill-suited to making an accurate prediction when the conditioning is completely changed at each diffusion step. Meaningful policies are still learnt when we ablate the action clipping (No Clipping), however the performance is worse, especially for Walker2d. The plots in Appendix B.4 show that without the action clipping, the action distributions produced are more heavy-tailed. Conditioning the policy on the noisy states during diffusion, rather than the denoised prediction (Noisy State Conditioning), also results in worse performance. Additionally updating the states during diffusion according to Equation 8 results in comparable performance to the base PolyGRAD algorithm. However, this modification increases the computation time required to about 80 hours for 1M environment steps (compared to 54 hours for the base algorithm), as it requires backpropagation through the policy at each diffusion step.

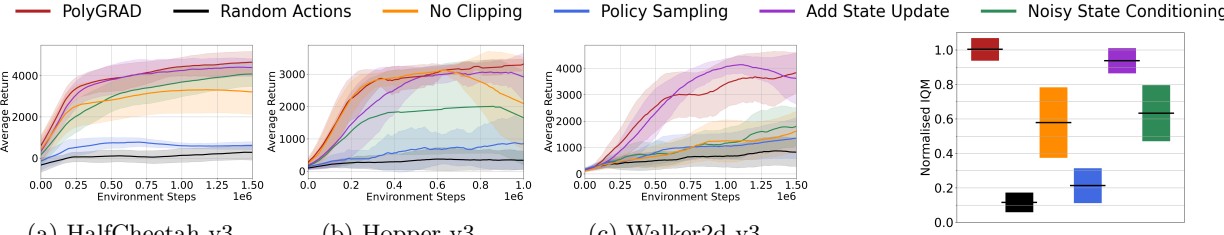

(a) HalfCheetah-v3    (b) Hopper-v3    (c) Walker2d-v3

Figure 8: Reward curves for PolyGRAD, and ablations and modifications to PolyGRAD. Shaded regions indicate standard deviation across 5 seeds.

Figure 9: 95% CIs of interquartile mean of normalised final performance aggregated across envs.

## 6    Discussion

**Limitations**   There are two key limitations of our approach. The first is that to maintain training stability, we found that it was necessary to update the policy slowly to avoid incorrect action distributions being generated. This likely increases the number of RL training updates required to achieve good performance. The second is that PolyGRAD struggles to achieve the correct action distribution if the policy has very low entropy. This may prevent PolyGRAD from achieving strong RL performance on some domains.

**Future Work**   To improve the stability issues mentioned above, we plan to investigate alternative algorithms for diffusing on-policy trajectories, with the aim of finding an algorithm that more reliably generates the correct action distribution. We would like to scale PolyGRAD to more complex environments such as image-based environments by utilising latent diffusion (Rombach et al., 2022). We would also like to test PolyGRAD on non-Markovian environments. PolyGRAD is well-suited to non-Markovian predictions due to the fact that it is trained to predict entire trajectories, rather than making single step predictions under a Markovian assumption. Another direction for future work is to perform a theoretical analysis of the

convergence properties of PolyGRAD to better understand if there are conditions under which it is guaranteed to converge to on-policy trajectories.

Finally, and perhaps most importantly, we would like to investigate whether there are situations in which PolyGRAD obtains better prediction errors at long horizons compared to existing autoregressive baselines. The results in Section 5.2 show that while PolyGRAD obtains strong performance when trained and evaluated on short trajectories ($h = 10$), performance deteriorates relative to baselines for longer trajectories. We hypothesise that PolyGRAD may be more robust than autoregressive models when trained on small datasets, where the predictions of single-step autoregressive models may be prone to quickly leaving the data distribution, resulting in erroneous predictions. Another line of investigation could be to train the denoising network on short trajectories, and use it to generate longer trajectories. Perhaps this may result in stronger performance and generalisation to longer trajectories. A final approach could be to investigate whether latent diffusion is better suited to accurate generation of long trajectories.

**Conclusion**   We have presented PolyGRAD, the first world model that can generate on-policy trajectories without autoregressive sampling. To enable this, instead of sampling actions directly from the policy, PolyGRAD gradually updates the trajectory of states and actions using diffusion. We have shown that PolyGRAD generates a good approximation to the correct on-policy action distribution, and therefore enables on-policy RL in imagination. PolyGRAD achieves solid performance in terms of trajectory prediction accuracy compared to state-of-the-art autoregressive baselines. Thus, PolyGRAD introduces a promising new paradigm for world modelling, with many possible directions for extensions and improvements in future work.

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

## A   Implementation Details

**Policy Parameterisation**   To make it easier to analyse the behaviour of PolyGRAD with policies of different entropy levels, we defined the policy so that the standard deviation of the policy is controlled by a single learnable parameter $\sigma_\phi$ which is independent of the state. Therefore, the Gaussian policies that we consider are of the form $\pi_\phi(a|s) = \mathcal{N}(\mu_\phi(s), \sigma_\phi)$, where $\phi$ indicates the parameters of an MLP. This policy parameterisation is commonly used in other deep RL implementations (Kostrikov, 2018).

**Initial State Conditioning**   We generated branched synthetic rollouts from initial states within the dataset. To achieve this, we sample the initial state uniformly at random from the dataset. To condition trajectory generation on the initial state, we perform inpainting by replacing the initial state in the trajectory with the sampled initial state during both training and inference. This is the same procedure used by Janner et al. (2022).

**Action Clipping During Diffusion**   During diffusion, we clip the action sequence so that it is within three standard deviations of the mean action output by the policy. Therefore, the update in Line 10 of Algorithm 2 is implemented as:

$$\widehat{\boldsymbol{\tau}}_{i-1}^a \leftarrow \text{clip}\big(\widehat{\boldsymbol{\tau}}_i^a + \delta \cdot \nabla_{\widehat{\boldsymbol{\tau}}_i^a} \log \pi_\phi(\widehat{\boldsymbol{\tau}}_i^a \mid \widehat{\boldsymbol{\tau}}_0^s), \min = \mu_\phi(\widehat{\boldsymbol{\tau}}_0^s) - 3\sigma_\phi, \max = \mu_\phi(\widehat{\boldsymbol{\tau}}_0^s) + 3\sigma_\phi\big) + \sqrt{\beta_i}\mathbf{z} \qquad (11)$$

**Imagined RL Training**   For imagined RL training in Algorithm 3, we used Advantage Actor Critic (A2C) (Mnih et al., 2016) with Generalised Advantage Estimation (GAE) (Schulman et al., 2016). We performed one update to the policy and value function for every four steps of data collection in the real environment. During each A2C training update, we generated 1024 synthetic on-policy rollouts of length 10. We restricted the minimum standard deviation of the policy, $\sigma_\phi$, to be 0.1.

We observed that RL training with PolyGRAD world models is unstable when there are large updates to the policy: if the policy is changed by a large amount, PolyGRAD may not consistently produce the correct action distribution. To address this, we defined the update magnitude as the average change in log-likelihood $|\log \pi_{\text{new}}(a|s) - \log \pi_{\text{old}}(a|s)|$ across all state-action pairs in the training batch. For each update to the policy, we tuned the learning rate via a linesearch so that the average update magnitude was within 20% of the target update magnitude, $\text{target}_{\Delta \log \pi}$. The hyperparameters used for A2C are shown in Table 1.

Table 1: A2C Hyperparameters

| Parameter | Value |
|---|---|
| Number of imagined trajectories per update | 1024 |
| Imagined trajectory length, $h$ | 10 |
| Generalised advantage estimation $\lambda$ | 0.9 |
| Critic learning rate | 3e-4 |
| Optimiser | Adam |
| Discount factor, $\gamma$ | 0.99 |
| Target policy update, $\text{target}_{\Delta \log \pi}$ | 0.01 |
| Entropy bonus weight | 1e-5 |
| Minimum policy std. dev. | 0.1 |
| Training steps per environment steps | 0.25 |

**Denoising Model**   The hyperparameters used for the denoising networks and diffusion process are summarised in Tables 2, 3 and 4. The noise schedule $\beta_i$ is defined using a cosine noise schedule (Nichol & Dhariwal, 2021). We used the implementation of a cosine noise schedule defined in Algorithm 1 of Chen (2023). We used the default parameter of $\tau_{\text{noise}} = 1$ to control the noise schedule, with the exception that for Hopper-v3 with the residual MLP denoising network we used $\tau_{\text{noise}} = 0.1$ (which reduces the noise level throughout the early steps of the forward process) as we found this obtained better prediction errors.

For the denoising network, we considered both a Residual MLP and a Transformer architecture, both trained to minimise the L2 loss. The Residual MLP has skip connections at each layer. To add conditioning on the step of the diffusion process, we add a learnable embedding of the diffusion step, $i$. Therefore, each layer has the form:

$$x_{L+1} = \text{linear}(\text{activation}(x_L)) + \mathrm{x}_L + \text{embed}(i)$$

To add action conditioning, we concatenate the actions to the inputs so that the input size is $(|S| + |A|) \times h$ where $h$ is the length of the trajectory. The final layer projects the output to the correct dimensionality of $|S| \times h$.

In the transformer denoising network, we first embed each noisy state and action pair using a learnt embedding function. The context length is equal to the length of the trajectory, $h$. We add conditioning upon the timestep in the diffusion process and the timestep in the trajectory by adding a learned embedding of each. Each transformer block consists of a LayerNorm followed by a multi-head causal self-attention layer and a 2-layer MLP. The linear output layer projects each of the $h$ embeddings to the correct dimensionality of $|S|$.

Table 2: Residual MLP Denoising Network Hyperparameters

| Parameter | Value |
|---|---|
| MLP Width | 1024 ($h = 10$) or 2048 ($h > 10$) |
| Batch size | 256 |
| Number of layers | 6 |
| Learning rate | 3e-4 |
| Optimiser | Adam |
| Training steps per environment steps | 1 |

Table 3: Transformer Denoising Network Hyperparameters

| Parameter | Value |
|---|---|
| Embedding dimension | 312 |
| Batch size | 256 |
| Number of layers | 6 |
| Self-attention heads | 4 |
| Learning rate | 1e-4 |
| Optimiser | Adam |
| Training steps per environment steps | 1 |

Table 4: Diffusion Hyperparameters

| Parameter | Value |
|---|---|
| Number of diffusion steps, $N$ | 128 |
| Noise schedule | cosine |
| Noise schedule parameter $\tau_{\text{noise}}$ | 1 (0.1 for Hopper-v3 + residual MLP) |

**Hyperparameter Tuning**  We reduced the cosine noise schedule parameter $\tau_{\text{noise}}$ to 0.1 for Hopper-v3 with the Residual MLP denoising network as we found this obtained better prediction errors. We used a larger MLP width of 2048 for the Residual MLP when the trajectory length was greater than 10. All other hyperparameters were kept constant across all environments. We found that the most critical hyperparameter for RL training was the size of the target policy update, $\text{target}_{\Delta \log \pi}$. To tune this parameter, we started with a large value and decreased it until RL training was stable for all MuJoCo environments.

# B    Additional Results

## B.1    Prediction Error Plots with Transformer and MLP Denoising Networks

Figure 10 compares the trajectory errors for PolyGRAD between the transformer denoising network and the residual MLP denoising network. We observe that for short trajectories ($h = 10$), similar prediction errors are obtained. However, for the longer trajectories the transformer denoising network obtains considerably better performance.

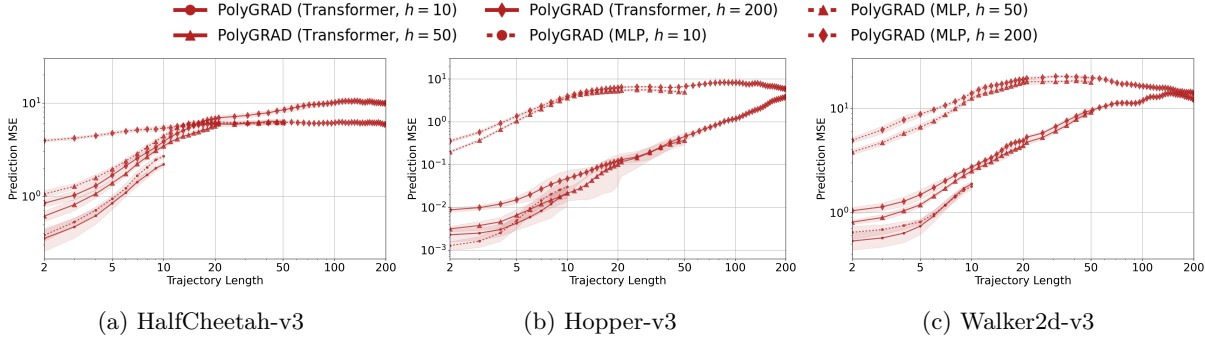

|     | (a) HalfCheetah-v3 | (b) Hopper-v3 | (c) Walker2d-v3 |

Figure 10: Plots of mean squared error (MSE) of predicted states vs ground truth states for PolyGRAD using either a transformer or residual MLP denoising network. Shaded regions indicate standard deviations over 5 seeds.

## B.2    Confidence Intervals for Mean Performance

In the main body of the paper, we reported 95% confidence intervals for the normalised interquartile mean (IQM) generated using the *rliable* framework (Figures 7 and 9). To provide a more complete picture, here we provide equivalent confidence intervals for the mean, rather than the IQM. We observe that the confidence intervals for mean are similar to those for IQM. This indicates that there are not many outliers in the results, as the mean is effected by outliers while IQM is not.

The intervals in Figure 12 compare PolyGRAD against other model-based and model-free algorithms. We observe that PolyGRAD outperforms the model-free algorithms for mean performance, but does not perform as well as Dreamer-v3. Figure 14 compares the base PolyGRAD algorithm against each ablation and variant of the algorithm. PolyGRAD significatly outperforms each variant/ablation except for Add State Update.

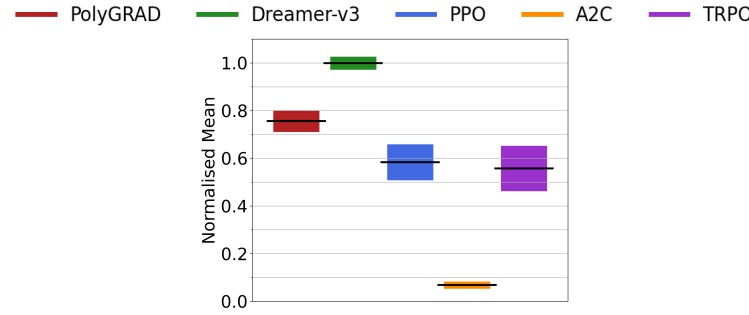

Figure 12: 95% confidence intervals of mean of normalised final performance aggregated across environments.

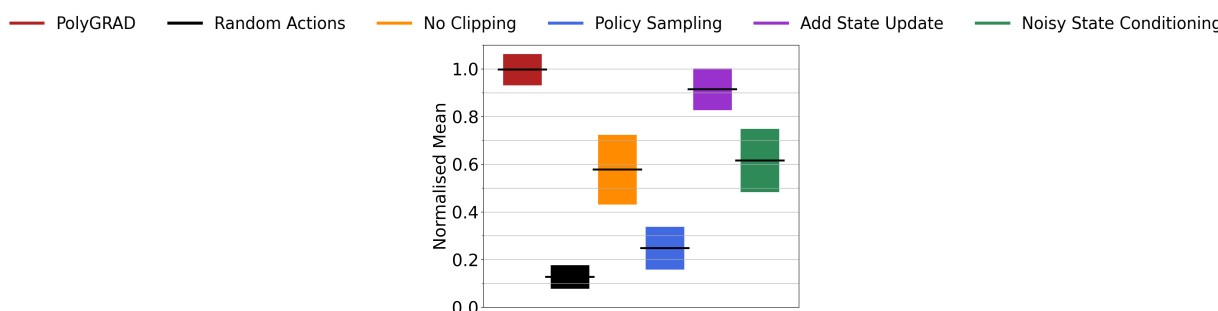

Figure 14: 95% confidence intervals of mean of normalised final performance aggregated across environments for PolyGRAD and each variant.

## B.3 Extended Plots of PolyGRAD Action Distributions for All Environments

Figures 15−17 show plots of action distributions produced by PolyGRAD in each of the MuJoCo environments with $h = 10$. Blue line illustrates the distribution of $a - \mu_\phi(s)$ for a batch of synthetic data. Each subplot is for a policy with a different entropy level. Dashed black line indicates the action distribution output by the policy.

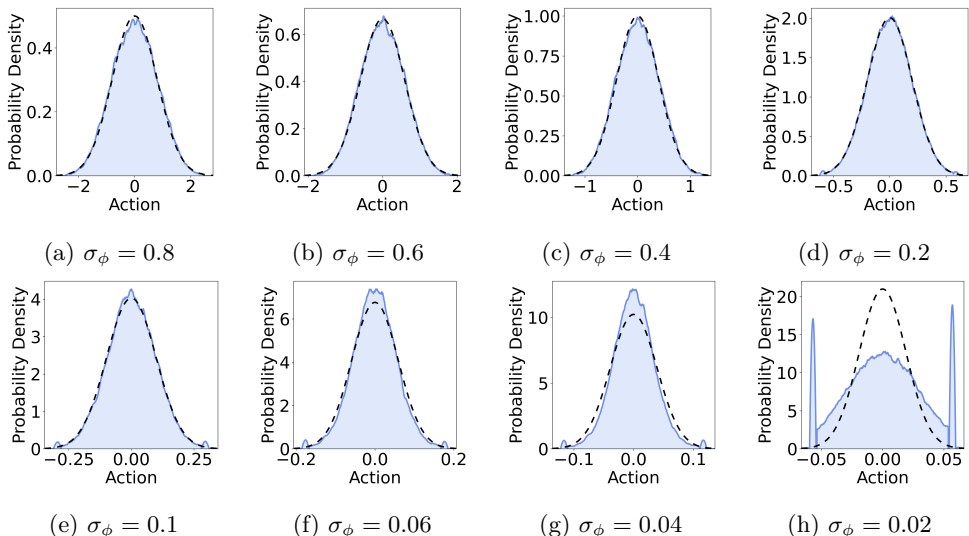

Figure 15: Walker2d-v3 action distribution plots for trajectories generated by PolyGRAD for $h = 10$.

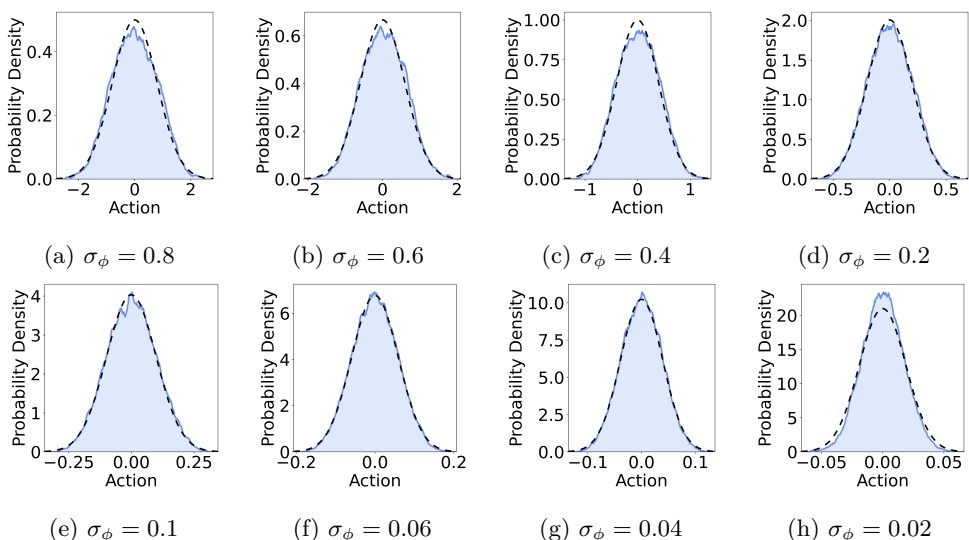

Figure 16: Hopper-v3 action distribution plots for trajectories generated by PolyGRAD for $h = 10$.

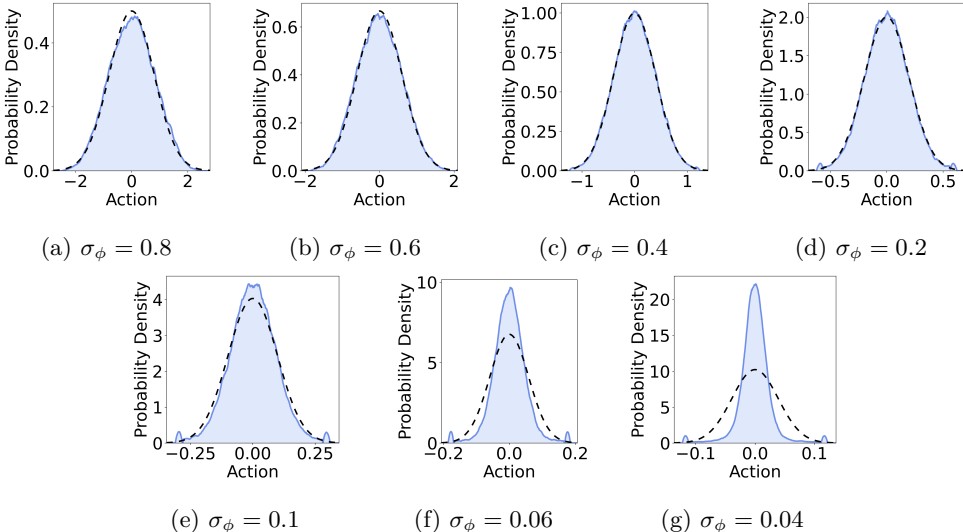

Figure 17: HalfCheetah-v3 action distribution plots for trajectories generated by PolyGRAD for $h = 10$. NaN values were produced before the minimum policy standard deviation of 0.02 was reached.

### B.4 PolyGRAD Action Distributions with Action Clipping Ablated

Figure 18 shows plots of the action distributions in Walker2d when the action clipping during diffusion is removed. Blue line illustrates the distribution of $a - \mu_\phi(s)$ for a batch of synthetic data and dashed black line is the policy distribution. We observe that when the action clipping is ablated, PolyGRAD tends to produce a heavy-tailed action distribution with some actions very far from the mean. This means that it still obtains the same standard deviation over actions as the policy, but the distribution over actions is no longer Gaussain. Thus, PolyGRAD is less effective at produce the correct action distribution when the action clipping is removed.

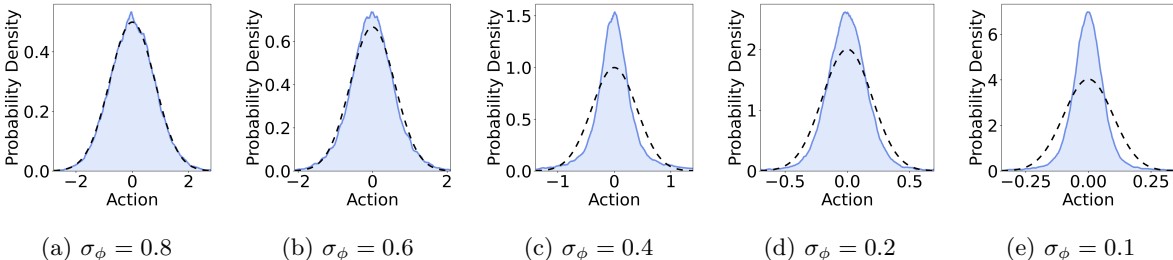

(a) $\sigma_\phi = 0.8$ (b) $\sigma_\phi = 0.6$ (c) $\sigma_\phi = 0.4$ (d) $\sigma_\phi = 0.2$ (e) $\sigma_\phi = 0.1$

Figure 18: Walker-v3 action distribution plots when the action clipping has been ablated. Without the action clipping, PolyGRAD tends to produce a heavy-tailed action distribution with some actions very far from the mean.

## C  Additional Experiment Details

### C.1  Experimental Setup

**Error Evaluation Experiments**   To collect the datasets, we ran Algorithm 3 until we had collected 1M transitions in each environment. Each world model was then trained on the same 1M transitions collected. The final policy produced by Algorithm 3 was used as the policy for sampling actions in each world model.

To generate synthetic rollouts, we sampled 500 initial states uniformly at random from the dataset and generated a synthetic rollout starting from each of these initial states. We computed the average mean squared error of the prediction across all 500 rollouts for each possible horizon length. We continued to train each world model until it obtained the best prediction error evaluation at a 5 step horizon, up to a maximum of either 1M gradient steps or 72 hours of training on an RTX 3090 GPU.

### C.2  Results Processing

**Smoothing of Reward Curves**   To generate the reward curves in Figures 6 and 8, the average return over 10 episodes of the current policy is evaluated after every 10,000 environment steps. To generate the plots we perform smoothing by computing a moving average with a window width of 10 evaluations.

**Confidence Intervals**   Figures 7 and 9 present 95% confidence intervals of the interquartile mean (IQM) of the normalised final performance aggregated across all three environments. Figures 12 and 14 are equivalent plots for the mean rather than IQM. To generate the confidence intervals, we use the *rliable* framework (Agarwal et al., 2021) which is designed to compute confidence intervals with limited seeds by aggregating the runs across all environments, and generating the confidence intervals using stratified bootstrapping. IQM is the performance on the middle 50% of combined runs, and is the metric recommended by rliable as it is robust to outliers (unlike the mean) and is more stastically efficient than the median.

To generate the confidence intervals, we first aggregate the final smoothed rewards obtained at the maximum number of environment steps for each run of each algorithm in Figures 6 and 8. Following Agarwal et al. (2021), we first normalise the rewards in each environment. We normalise the rewards by dividing by the best average final total reward obtained by any algorithm for each environment. We then use the normalised final rewards for each run to compute the aggregated IQM confidence intervals using the rliable package with 50,000 bootstrapping repeats.

### C.3  Baselines

**Probabilistic MLP Ensemble**   We train an ensemble of MLPs that output a Gaussian distribution over the next state. Following Janner et al. (2019) and  Yu et al. (2020), we train an ensemble of 7 MLPs with 4 layers and 200 hidden units per layer. Each model outputs the mean and variance over the next state using a two-head architecture. When sampling from the model, we sample uniformly from the 5 MLPs that obtain

the lowest prediction loss on a held-out test set. We use the PyTorch implementation publicly available at github.com/Xingyu-Lin/mbpo_pytorch.

**Transformer** We use the same transformer architecture and hyperparameters as for the PolyGRAD transformer denoising model described in Appendix A. Like the denoising model, the transformer is trained using the L2 loss. However, instead of the denoising objective, the transformer is trained to predict the next state given the context of a sequence of previous states and actions. We use a maximum context length of 15 state-action pairs. To generate trajectories, we query the transformer autoregressively.

**Autoregressive Diffusion** We use the same diffusion model as Synther (Lu et al., 2023), except that we add action conditioning by concatenating the action to the input. This diffusion model is used to make a one-step prediction of the next state, conditioned on the current action. To generate synthetic trajectories, we sample actions from the policy and autoregressively generate one step of the trajectory at a time by completing the reverse diffusion process individually for each step.

**Dreamer-v3** We use the implementation of Dreamer-v3 (Hafner et al., 2023) available at github.com/NM512/dreamerv3-torch. For RL training, we perform one step of RL training by directly performing backpropagation through the dynamics model for every four steps of data collection in the environment. All hyperparameters are set to the defaults used by Dreamer-v3 for MuJoCo. To evaluate the prediction errors, we first initialise the latent state by observing a sequence of five states and actions from the real environment. We then perform open-loop predictions by predicting the next latent states and decoding these using the decoder.

**Model-Free RL Algorithms** We use the implementations of PPO, TRPO, and A2C from Stable Baselines 3 (Raffin et al., 2019). For PPO and TRPO we used the default hyperparameters. For A2C we used the default hyperparameters, with the exception that we set the learning rate to $3e-4$ and the entropy coefficient to $1e-5$ as we found this worked better for the MuJoCo benchmarks than the default parameters.

### C.4 Ablations and Modifications

**Random Actions**: We sample random actions at the beginning of the diffusion process from a standard normal distribution, and they are not updated during diffusion.

**Policy Sampling**: During each step of the diffusion process, we obtain new actions by directly sampling from the policy action distribution $\pi_\phi(\cdot \mid \widehat{\boldsymbol{\tau}}_0^s)$.

**No Clipping**: The actions are not clipped during the diffusion process.

**Add State Update**: In addition to updating the actions according to the score of the policy distribution, we also update the states to increase the likelihood of the actions. For this modification, we first compute the noise prediction per Line 7 of Algorithm 2. We use this to compute the estimate of the denoised states, $\widehat{\boldsymbol{\tau}}_0^s$:

$$\widehat{\boldsymbol{\tau}}_0^s \leftarrow \frac{1}{\sqrt{\alpha_i}} \cdot \widehat{\boldsymbol{\tau}}_i^s - \frac{\sqrt{1-\alpha_i}}{\sqrt{\alpha_i}} \cdot \widehat{\boldsymbol{\epsilon}} \tag{12}$$

We then update the estimate of the denoised states according to by using the score of the policy distribution with respect to the states:

$$\widehat{\boldsymbol{\tau}}_0^s \leftarrow \widehat{\boldsymbol{\tau}}_0^s + \delta \cdot \nabla_{\widehat{\boldsymbol{\tau}}_0^s} \log \pi_\phi(\widehat{\boldsymbol{\tau}}_i^a \mid \widehat{\boldsymbol{\tau}}_0^s)$$

Computing this gradient requires performing backpropagation through the policy. By rearranging Equation 12 we then compute an updated noise prediction based on the modified denoised state prediction

$$\widehat{\boldsymbol{\epsilon}} \leftarrow \frac{1}{\sqrt{1-\alpha_i}} \widehat{\boldsymbol{\tau}}_i^s - \frac{\sqrt{\alpha_i}}{\sqrt{1-\alpha_i}} \widehat{\boldsymbol{\tau}}_0^s$$

This final noise prediction is then used to update the states using the diffusion update in Line 13 of Algorithm 2.

**Noisy State Conditioning**: Instead of conditioning the policy on the denoised state prediction ($\widehat{\boldsymbol{\tau}}_0^a$ in Line 10 of Algorithm 2) we condition the policy on the noisy states ($\widehat{\boldsymbol{\tau}}_i^a$).

### C.5 Computational Requirements

For PolyGRAD, one run of RL training up to 1M environment steps (Algorithm 3) requires 54 hours of computation time on an RTX 3090 GPU.

