# OpenReview forum: "World Models via Policy-Guided Trajectory Diffusion"
_TMLR — Accepted by TMLR_

### Review · Reviewer_Rzwm · 2024-01-08

**Summary Of Contributions:**

This paper introduces Policy-Guided Trajectory Diffusion, an approach that uses a world model to perform on-policy RL on trajectories “in imagination”. Crucially, the proposed approach does not rely on autoregressive sampling but denoises full trajectories. To keep the action compatible with the current policy the algorithm updates them using the policy score function.

The proposed method is evaluated on 3 simple continuous control environments and appears to outperform on-policy RL baselines but not DreamerV3, a strong world model baseline. Furthermore the paper presents a number of ablations.

**Audience:**

Yes

**Claims And Evidence:**

No

**Requested Changes:**

* page 6: I think there’s a log missing in line 2
* Consider citing Schubert et al. "A Generalist Dynamics Model for Control." arXiv preprint arXiv:2305.10912 (2023) as another recent paper on transformer dynamics models.
* It is claimed that PolyGrad outperforms on-policy RL. However from the figure this could be due to statistical noise. Could you include a test?
* Not sure if Welling & Teh is the right reference for Langevin dynamics, since that paper is about using it with stochastic gradients.
* Give smoothing details in figures in e.g. figure 6.
* Please give intuition for why eq (10) makes sense. I understand that there should be a fixed point but why this update?
* Why A2C? Why not PPO as a base algorithm?
* I’m not sure the paragraph on diffusion augmentation methods in the related works section needs to be there. These papers are quite different.

**Strengths And Weaknesses:**

Strengths:
* interesting, novel formulation with nice proof-of-concept results
* Interesting discussion of theoretical connections of the proposed approach to Langevin dynamics and classifier guidance.

Weaknesses:
* The paper claims scalability but only evaluates on 3 simple environments. I think the claims in the paper should be tempered a bit.
* Typically world models are motivated by a desire for more sample efficiency. In that context the on-policy baselines are not very informative. A comparison with more model-based methods or off-policy RL algorithms would be more meaningful.
* No improvement on DreamerV3

---

> ### Author Response · Authors · 2024-01-22
> **Response to Reviewer Rzwm**
>
> Thank you for the time you have spent reviewing our paper! We will respond to each of your requested changes and comments below.
>
> &nbsp;
>
> ## Requested Changes
>
> > Scalability claims
>
> We have removed the mention of scalability from the abstract, introduction, and conclusion. In the future work section, we have added that we would like to scale PolyGRAD to more complex environments in future work.
>
> &nbsp;
> > page 6: there’s a log missing in line 2
>
> Thanks for pointing this it out, it is now fixed.
>
> &nbsp;
> > Consider citing Schubert et al.
>
> Thank you for this suggestion, we have added this reference to the related work.
>
> &nbsp;
> > It is claimed that PolyGrad outperforms on-policy RL. However from the figure this could be due to statistical noise. Could you include a test?
>
> We have generated 95\% confidence intervals for these results.
> To do this using limited seeds, we used the *rliable* framework (Agarwal et al. 2021), which normalises the results in each environment, and aggregates the results across all environments to perform statistical testing using stratified bootstrapping.
> Extended details for how the confidence intervals are generated are now provided in Appendix C.2.
>
> Figures 7 and 9 in the revised paper show 95\% confidence intervals of the interquartile mean of the normalised performance at the end of training (i.e. the average performance on the middle 50\% of runs). The confidence intervals show that PolyGRAD obtains statistically significant improved final performance over the model-free RL algorithms PPO, TRPO, and A2C at the end of training.
>
> *Rishabh Agarwal et al. Deep reinforcement learning at the edge of the statistical precipice. Advances in Neural Information Processing
> Systems, 2021.*
>
> &nbsp;
> > Not sure if Welling & Teh is the right reference for Langevin dynamics, since that paper is about using it with stochastic gradients.
>
> Thank you for pointing out that this isn't the most suitable reference.  We have changed the citation to Rossky et al. 1978 as we have found is the first work to introduce Langevin dynamics as a method for Markov chain Monte Carlo sampling. We have also added a reference to Chapter 5 of Neal 2011 as a standard textbook reference on Langevin dynamics for the interested reader.
>
> *Peter J Rossky, Jimmie D Doll, and Harold L Friedman. Brownian dynamics as smart monte carlo simulation.
> The Journal of Chemical Physics, 69(10):4628–4633, 1978.*
>
> *Radford Neal. MCMC using Hamiltonian dynamics. In Handbook of Markov Chain Monte Carlo, chapter 5.
> Chapman and Hall / CRC Press, 2011.*
>
> &nbsp;
> > Give smoothing details in figures in e.g. figure 6.
>
> We have provided these details in Appendix C.2. The details given in the appendix are the following:
>
> *To generate the reward curves in Figures 6 and 8, the average return over 10 episodes of the current policy is evaluated after every 10,000 environment steps. To generate the
> plots we perform smoothing by computing a moving average with a window width of 10 evaluations.*
>
> &nbsp;
> > Please give intuition for why eq (10) makes sense.
>
> We have modified the text following Equation 10 to give intuition for the update. The text following Equation 10 now reads:
>
> *The intuition for Equation 10 is as follows. If the policy guidance is too strong,
> all of the actions will be guided to be very near to the mean of the policy distribution. Therefore, the
> standardised actions will have a variance that is too low (i.e. below 1) and Equation 10 will reduce the action
> update scale. Likewise, if the guidance is too weak, the actions will remain spread out far from the policy
> mean. Thus, the standardised actions will have a variance that is too high and Equation 10 will increase the
> action update scale.*
>
> &nbsp;
> > Why A2C? Why not PPO as a base algorithm?
>
> The advantage clipping in PPO ensures that the change in action distribution is small, even when making multiple policy gradient updates using the same batch of data. In our work, because we have a world model we can generate an unlimited amount of data without interacting with the environment so we have no need to make multiple updates using the same data. Since we only make one update per batch, the clipping in the PPO update has no effect (as the likelihood ratio of the new and old policy is always 1 for the first update). For this reason, we decided to use A2C. See here for a similar discussion from the Dreamer-v2 authors: https://github.com/danijar/dreamerv2/issues/2.
>
> However, we still think it could be worthile investigating the use of PPO instead of A2C.  This is because the main computational bottleneck of our approach is the time required to run diffusion to generate the synthetic data for each policy gradient update. If we performed multiple updates with each batch of synthetic data (using PPO to enable this) this might dramatically reduce the computation time required by our approach. So this could be a good direction for future work.

---

> > ### Author Response · Authors · 2024-01-22
> > **Response continued**
> >
> > > I’m not sure the paragraph on diffusion augmentation methods in the related works section needs to be there
> >
> > Thanks for this suggestion, we agree that diffusion augmentation is less relevant. We have removed this from the related work, and added the concurrent paper "Policy-Guided Diffusion" to the related work as suggested by Reviewer ZEpn.

---

### Review · Reviewer_ZEpn · 2024-01-10

**Summary Of Contributions:**

The paper proposes to use a policy to condition a diffusion
process for generating state sequences in an MDP.
This is an advantageous way of sampling trajectories from a
policy in an MDP that combines the capabilities of a diffusion
model to generate a temporally and spatially coherent trajectory
with the space of feasible trajectories induced by some policy.
The main methodological and algorithmic contributions are in
Section 4 for training/sampling from the policy-conditioned models.
Then Section 5.1 evaluates the ability of sampling from synthetic
trajectories for matching Gaussians. Section 5.2 compares
methods for MuJoCo environments (HalfCheetah, Hopper, Walker2d)
in terms of MSE of the predictions. Section 5.3 uses the
sampled trajectories for policy learning,. And Section 5.4
ablates design choices.

**Audience:**

Yes

**Claims And Evidence:**

Yes

**Requested Changes:**

The paper seems to have been done concurrently with the paper
[Policy-Guided Diffusion](https://www.robot-learning.ml/2023/files/paper13.pdf),
presented at the NeurIPS 2023 workshop on robot learning.
I see this as an indication of the relevant of the idea of
policy-guided diffusion and believe it would be helpful to
connect to their methodology and experiments in the final version of the paper.

**Strengths And Weaknesses:**

# Strengths
1. This is a timely paper that to the best of my knowledge has
   a reasonable and sound experimental evaluation.
   Figure 3 clearly demonstrates the ability of fitting
   the synthetic data.
   Figure 5 clearly demonstrates the ability of modeling
   real systems and how will it compares to existing
   methods as the generated trajectory lengths are increased,
   and Figure 6 seems like a reasonable attempt to document on
   using the generations for policy learning.
2. The idea of policy conditioning is reasonable and
   clearly motivated throughout the main paper, and the
   connection to classifier guidance makes a lot of sense.

# Weaknesses
I do not have any significant concerns with this paper
--- it is a well-documented idea and investigation.
My one concern is that it is difficult to evaluate the contribution in comparison
to existing approaches as there are no standardized benchmarks
for a) learning dynamics models and b) learning policy-conditioned
models.
For example, while the experimental results in Figure 5
compare to other methods on MuJoCo experiments, all of the
baselines are newly reported in this paper and not directly comparable
to any existing published results in the literature.
The paper seems to do a reasonable job at properly describing
and setting up the experimental settings, so I do not see
  this as a strong issue for this paper

---

> ### Author Response · Authors · 2024-01-22
> **Response to Reviewer ZEpn**
>
> Thank you for the time you have spent reviewing our paper!
>
> We have updated the related work section to include a thorough comparison to the concurrent work by Jackson et al. (at the top of page 4). This part of the related work section now reads as follows:
>
> *"Also concurrently, and most related to our work, Jackson et al. (2023) guide diffusion with a policy to increase the likelihood of generated trajectories under the policy, in a similar manner to PolyGRAD. Jackson et al. (2023) use this approach to generate synthetic datasets for offline, off-policy RL. In contrast, PolyGRAD generates on-policy trajectories for online, on-policy RL in imagination. Unlike Jackson et al. (2023), we also analyse the connection between PolyGRAD, classifier-guidance, and score-based generative models. Furthermore, we show that the scale of the action guidance can be tuned automatically online to approximately generate the correct on-policy action distribution."*

---

### Review · Reviewer_bqnZ · 2024-01-17

**Summary Of Contributions:**

This paper addresses the problem of learning world models for model-based RL. Differently from previous methods, which are generally autoregressive and generate trajectories one step at a time, the paper proposes to generate full trajectories (state-action-reward sequences) in one-pass through diffusion. The key technical hurdle to adapt diffusion to this setting is to make the model generate on-policy data, i.e., to generate actions that are consistent with the policy conditioned on the generated states. In the paper, the latter is overcame by training a denoising model for state sequences (conditioned on actions sequences) and then interleaving state sequence generations and action sequence updates in the directions of the policy score. The method is plugged into an on-policy policy optimization algorithm and evaluated on various Mujoco tasks (on proprioceptive states).

**Audience:**

Yes

**Broader Impact Concerns:**

This paper fits into the category of fundamental research, as all the experiments are in simulation and directly applying the proposed method in the real-world seems non-trivial.

**Claims And Evidence:**

No

**Requested Changes:**

I think that the changes I am mentioning below can improve the current submission, although a more convincing empirical analysis is what could dramatically improve the value of the paper.
- I would suggest the author to drop the claim that generating trajectories in one pass improves the accuracy over autoregressive models, which is not fully supported by the experiments, and rework the narrative accordingly;
- I would refrain to say that Algorithm 3 outperforms PPO/TRPO, as the statistical significance of the improvement is unclear from the given experiments;
- I would suggest the author to at least explain how the approach can be employed for planning other than policy optimization. Having experiments on planning would be great, but at least discussing how that shall work is already valuable.

**Strengths And Weaknesses:**

As a disclaimer, I am coming from the RL field with very limited knowledge of generative models and diffusion especially. Thus, I cannot fully judge the novelty with respect to related works, or whether the technical solutions to adapt diffusion to on-policy sampling is really substantial. Having said that, from an RL perspective the paper seems to introduce interesting ideas, although the experiments do not fully support the claim that generating long-horizon trajectories in one diffusion pass benefits accuracy w.r.t. autoregressive models of prior works. I summarize below what are, in my opinion, the main pros and cons of the paper. Then, I report some more detailed comments.

*STRENGTHS*
- How to limit the error propagation across long-horizon trajectories looks like a relevant and important direction;
- The paper is very well written and easy to follow;
- The empirical analysis is thorough in terms of validation and ablations;
- The proposed world modelling algorithm can work in non-Markovian environments (including POMDPs?)
- The discussion section at the end is upfront in outlining how the method could be improved and interesting directions.

*WEAKNESSES*
- Experiments does not seem to fully back the claim that generating trajectories in one pass improves over autoregressive models over long horizons;
- Theoretical analysis is left as future work;
- The paper does not explore the use of the trained world model for planning;
- Statistical significance of the results is questionable, as very few seeds are considered (mostly 4) and confidence intervals are not reported (though it is hard to get sensible c.i. without adding more runs);
- Image generation is arguably the most popular application of diffusion, but the paper only consider physical states instead of learning from images.

*COMMENTS*
1) With the proposed method, how can we ensure that the model does not generate unfeasible trajectories? E.g., from my understanding the initial state s^0 is forced through, but how can we guarantee that the transition s^0 to s^1 remains feasible? The empirical validation consider (mostly) unconstrained domains, where the latter may be less of an issue.
2) Can the model be used for planning as is, or that would require some non-trivial modifications?
3) One drawback that is often mentioned about diffusion is the computational cost of doing inference. Can the authors comment on the computational complexity of the inference, especially w.r.t. with autoregressive models in prior works (computational times are reported in the experiments)? Do the authors believe that this may be a major issue for employing their method for doing planning?
4) In Algorithm 3, the generator of action sequences is trained at every iteration to keep the data on-policy, but the denoising network for state sequences is also re-trained. I guess this is crucial to avoid using the generator of action sequences out-of-distribution. However, I was wondering whether one could try to collect a dataset with a nice covering of  trajectories, then train the state-sequences generator once and only train the generator for actions during the loop (this may be a dumb idea as I am not knowledgeable, as I said before).
5) Another perhaps interesting question to ask empirically is whether PolyGRAD trained to generate short trajectories display some generalization when used to generate longer trajectories. Could this experiment be done technically?
6) The paper mentions that the method suffer from severe instability when paired with policy learning, which is mitigated with very slow policy updates. Providing a theoretical characterization of this behavior seems extremely interesting.

---

> ### Author Response · Authors · 2024-01-22
> **Response to Reviewer bqnZ**
>
> Thank you for the time you have spent reviewing our paper! We will first respond to your requested changes, followed by your comments.
>
> ## Requested Changes
> ___
>
> > I would suggest the author to drop the claim that generating trajectories in one pass improves the accuracy over autoregressive models
>
> We have reworked the paper the ensure that the claims match the experimental results as closely as possible. The abstract and introduction now state:
>
> *Our results demonstrate that PolyGRAD outperforms state-of-the-art baselines in terms of trajectory prediction error for **short** trajectories, with the **exception of autoregressive diffusion**. For short trajectories, PolyGRAD obtains similar errors to autoregressive diffusion, but with lower computational requirements. For **long trajectories**, PolyGRAD obtains **comparable performance to baselines.***
>
> The claim that PolyGRAD outperforms the baselines except for autoregressive diffusion for short trajectories is shown in Figure 5, where the line for PolyGRAD with *h=10* (i.e. trajectories of length 10) outperforms all of the baselines except for autoregressive diffusion. The claim that PolyGRAD obtains comparable performance to baselines for long trajectories is evidenced by the lines for *h=50* and *h=200* in Figure 5 which are comparable to most baselines in most of the environments.
>
> We have also extended the discussion of these results to the Future Work section in Section 6:
>
> *Finally, and perhaps most importantly, we would like to investigate whether there are situations in which PolyGRAD obtains better prediction errors at long horizons compared to existing autoregressive baselines. The results in Section 5.2 show that while PolyGRAD obtains strong performance when trained and evaluated on short trajectories (h = 10), performance deteriorates relative to baselines for longer trajectories. We hypothesise that PolyGRAD may be more robust than autoregressive models when trained on small datasets, where the predictions of single-step autoregressive models may be prone to quickly leaving the data distribution, resulting in erroneous predictions. Another line of investigation could be to train the denoising network on short trajectories, and use it to generate longer trajectories. Perhaps this may result in stronger performance and generalisation to longer trajectories. A final approach could be to investigate whether latent diffusion is better suited to accurate generation of long trajectories.*
>
> We hope you agree that this is a fair summary of the results, although please let us know if you think further revision is required.
>
> &nbsp;
>
> > I would refrain to say that Algorithm 3 outperforms PPO/TRPO, as the statistical significance of the improvement is unclear from the given experiments
>
> We have generated 95% confidence intervals for these results. To do this using limited seeds, we used the *rliable* framework (Agarwal et al. 2021), which normalises the results in each environment, and aggregates the results across all environments to perform statistical testing using stratified bootstrapping. Extended details for how the confidence intervals are generated are now provided in Appendix C.2.
>
> Figures 7 and 9 in the revised paper show 95% confidence intervals of the interquartile mean of the normalised performance at the end of training (i.e. the average final performance on the middle 50% of runs). The confidence intervals show that PolyGRAD obtains statistically significant improved final performance over the model-free RL algorithms PPO, TRPO, and A2C at the end of training.
>
> *Rishabh Agarwal et al. Deep reinforcement learning at the edge of the statistical precipice. Advances in Neural Information Processing Systems, 2021.*
>
> &nbsp;
>
> > I would suggest the author to at least explain how the approach can be employed for planning other than policy optimization
>
> Using diffusion models as a planner or trajectory optimiser (without learning a policy) is an existing line of work, initiated by the paper "Planning with diffusion for flexible behavior synthesis." Janner et al. ICML (2022). We have extended the discussion of this paper in the related work to make it clear that it addresses planning via diffusion models. Note that Janner et al. use guided diffusion as a trajectory optimiser for trajectory generation, so this is different to standard planning algorithms.
>
> The main novelty of our work is that it introduces policy guidance to enable entire synthetic on-policy trajectories to be generated with one pass through a diffusion model. This enables our approach to be viewed as a "world model" (as it simulates what would happen by executing a separate policy), thus enabling imagined on-policy RL (which arguably can be thought of as a type of planning).

---

> ### Author Response · Authors · 2024-01-22
> **Response continued**
>
> To our knowledge, most standard planning methods do not require on-policy trajectories. Model predictive control based on the cross-entropy method requires trajectories to be sampled, but does not require those trajectories to be on-policy. Monte Carlo tree search approaches require individual state transitions to be generated, not entire trajectories. Therefore, we do not think our approach is particularly applicable to either of these planning approaches. However, we do think that using guided diffusion to generate samples for the cross-entropy method is an interesting research idea.
>
> &nbsp;
>
> ## Comments
> ___
>
> > With the proposed method, how can we ensure that the model does not generate unfeasible trajectories?
>
> There is no constraint  that prevents the model from generating physically infeasible trajectories. We rely on the model being able to learn realistic transitions from the training data, such that the synthetic data generated is adequately accurate. This is the standard approach in model-based reinforcement learning with deep neural networks (e.g. Hafner et al. 2019, Hafner et al. 2021, Micheli et al. 2023).
>
> &nbsp;
>
> > Can the model be used for planning
>
> See the discussion in the last section of the "Requested changes".
>
> &nbsp;
>
> > One drawback that is often mentioned about diffusion is the computational cost of doing inference
>
> We found that diffusion indeed tends to be slower than most other methods, as shown in Figure 4 (although transformers are also quite slow). In our imagined RL setup, the computation cost is not a great concern. Our implementation takes approximately 0.5 seconds to generate a batch of data (10,000 transitions) for policy training. While this data generation is rather slow, we use the data to learn a feedforward policy which can then be executed very quickly at test time.
>
> For methods that run diffusion online at test time, computation time is indeed a concern. As discussed above, "Planning with diffusion for flexible behavior synthesis." Janner et al. ICML (2022) use a diffusion model as a trajectory optimiser/planner. The results in that paper show that the method requires ~0.2 seconds to select each action at test time, which might be prohibitively slow for many applications. Despite this, it has also become common to use diffusion models as policies, which also requires running diffusion at each step to generate each action (e.g. Pearce et al. 2023).
>
> &nbsp;
>
> > In Algorithm 3, the generator of action sequences is trained at every iteration to keep the data on-policy
>
> Note that the "generator of action sequences" is just the policy. So we train this at every iteration because the aim is to improve the policy. We continually keep training the state sequence denoising model so that we can continue to improve the trajectory predictions as we collect more data covering more of the state-action space.
>
> We agree that our approach could certainly be adapted to the offline RL setting. We could train the state denoising model once using a fixed batch of offline data, and then keep it fixed. Then, during training we would only update the policy to find a performant policy offline.
>
> An offline RL variant of our approach is an exciting direction for future work!
>
> &nbsp;
>
> > Another perhaps interesting question to ask empirically is whether PolyGRAD trained to generate short trajectories display some generalization when used to generate longer trajectories. Could this experiment be done technically?
>
> We agree that this is an interesting direction to investigate. The only technical constraint is that it must be necessary to train the denoising model on sequences of a different length to the sequences it is evaluated on. This can be achieved with most sequence models (e.g. transformers, 1D convolutions, RNN, etc.). We have added a discussion of this as a possibility to the future work section:
>
> *"Another line of investigation could be to train the denoising network on
> short trajectories, and use it to generate longer trajectories. Perhaps this may result in stronger performance
> and generalisation to longer trajectories."*
>
> &nbsp;
>
> > The paper mentions that the method suffer from severe instability when paired with policy learning, which is mitigated with very slow policy updates. Providing a theoretical characterization of this behavior seems extremely interesting.
>
> We agree that this is a very interesting problem. We have mentioned a theoretical analysis of our algorithm as a possible direction for future work in Section 4.2 as well as the future work section.

---

> > ### Comment · Reviewer_bqnZ · 2024-02-07
> >
> > I want to thank the authors for their detailed replies and for amending the paper following reviewers' suggestions.
> >
> > I have noticed this (very recent) related work on generating trajectories in a single-pass via diffusion "Diffusion World Model" https://arxiv.org/pdf/2402.03570.pdf. Perhaps this can be mentioned in a final version of the paper alongside other related works

---

> > > ### Author Response · Authors · 2024-02-08
> > > **Thank you for your response**
> > >
> > > Thanks for getting back to us! We agree that the paper is much improved following the reviewers' suggestions.
> > >
> > > Making sure the literature review is thorough is very important to us. We have updated the literature review to cite (Yang et al., 2023) as it was recently brought to our attention that this is another very recent work that uses autoregressive diffusion for world modeling.
> > >
> > > The paper "Diffusion World Model" was uploaded to arxiv almost two months following our submission. Furthermore, "Diffusion World Model" does not cite any of the four preceding papers on diffusion world models (Anonymous, 2023; Zhang et al. 2023; Yang et al. 2023; and our submission), all of which we cite in our paper. It does not even cite the previous paper called "Diffusion World Models" (with an "s"). For these reasons, we do not think it is reasonable for us to be expected to cite this new work.
> > >
> > > Thanks again for your help improving our paper, we really appreciate it!
> > >
> > > -The authors
> > > ___
> > >
> > >
> > > *Anonymous. Diffusion world models. OpenReview, 2023.*
> > >
> > > *Mengjiao Yang, Yilun Du, Kamyar Ghasemipour, Jonathan Tompson, Dale Schuurmans, and Pieter Abbeel.
> > > Learning interactive real-world simulators. arXiv preprint arXiv:2310.06114, 2023.*
> > >
> > > *Lunjun Zhang, Yuwen Xiong, Ze Yang, Sergio Casas, Rui Hu, and Raquel Urtasun. Learning unsupervised
> > > world models for autonomous driving via discrete diffusion. arXiv preprint arXiv:2311.01017, 2023.*

---

> > > > ### Comment · Reviewer_bqnZ · 2024-02-09
> > > >
> > > > Let me just clarify what I meant in the previous message: I was not requesting you to cite "Diffusion World Models", I was rather letting you know of its existence (so that you can decide yourself whether to mention it or not in the paper).
> > > >
> > > > However, let me add this. I can conceive two reasonable motivations for not citing a related paper: Either the authors are not aware of its existence (as it often happens, but doesn't apply here), or the authors think the reader would not benefit from knowing this other paper exists.
> > > >
> > > > Instead, not citing because it is a later work (what is the problem? You can of course mention that in your manuscript) or as a "retaliation" as this other paper does not mention prior works, do not sound like reasonable motivations.
> > > >
> > > > Best,
> > > > Reviewer bqnZ

---

> > > > > ### Author Response · Authors · 2024-02-09
> > > > > **A clarification**
> > > > >
> > > > > Thank you for the clarification.
> > > > >
> > > > > We agree with the points you made. But, to be clear, our choice to initially not cite this paper was not intended as a “retaliation”. Rather, it was a reflection of the fact that the omission of references to what we perceive to be all of the most relevant related works in this area gave us the impression that the Arxiv preprint  you shared was an early draft of a paper still under development. However, after reading the new paper more thoroughly we see that it  does propose interesting ideas that do warrant contextualising and discussing in our own work.
> > > > >
> > > > > We will include such a discussion in the final version of our paper.
> > > > >
> > > > > Thanks for taking the time to discuss this with us.
> > > > >
> > > > > Kind regards,
> > > > >
> > > > > The authors

---

### Author Response · Authors · 2024-01-22
**Thank you for your reviews**

Dear reviewers,

Thank you for the valuable time you have spent reviewing our submission.

We have uploaded a new version of our paper, and we will respond to each of your reviews separately.

Please let us know if there are any remaining issues that you would like to discuss!

---

### Decision · Action_Editor_Z6V9 · 2024-02-08

**Recommendation:** Accept with minor revision

**Comment:**

The paper is easy to read and the reviewers generally agreed on acceptance. The decision is Accept with Minor Revisions, as I would like to make two requests that I can check without further review.

Primarily, the number of runs should be consistent between methods. But some have 4 runs and others have 5. Please fix this, or explain why it is difficult to fix.
(And once you are fixing this, I highly encourage you to do more runs, even just 3-5 more runs since it will make results more significant).

Second, please include another figure like Figure 7, but showing 95% CIs for more than just the interquartile. It is not required that the outcome is significant in this case, but it provides a more complete picture, especially because it better matches the mean reported in learning curves. Also note that if it is significant, then you can claim that it is significant according to both metrics.

Minor question that does not have to be answered in the paper but I am curious about: why not also learn on real trajectories? Namely, learn on a mix of synthetic and real trajectories? Is there a complication with doing this that I am missing? Or did you find it performs more poorly?

**Audience:**

The TMLR audience will be interested in this work.

**Claims And Evidence:**

The claims are reasonably well supported by the evidence. The addition of significance testing, according to the guidelines from Rishabh et al, was an important improvement. However, it would be more clearly convincing if at least 5 runs were used consistently (in some cases only 4 runs were used for other methods, which is odd). And even better would be to also show Figure 7, using more than just the 50% of runs, say for example showing intervals over all the runs. I understand the IQM is more statistically robust, but it also shows a different conclusion. It would be helpful to report both. If there is only statistical significance in terms of IQM, that would be useful to know.

Of course, it is even easier to simply do a few more runs in these three environments. There is no hyperparameter sweeping, so it would not be expensive to do a few more runs.

---

> ### Author Response · Authors · 2024-02-24
> **Response to Action Editor**
>
> Dear Action Editor,
>
> Thank you for recommending acceptance of our paper.
>
> We have uploaded the final camera ready version with the following changes:
>
> - There are now 5 seeds for all algorithms in all experiments.
> - There are now confidence intervals for the mean (as well as interquartile mean (IQM)) computed using the rliable framework. The confidence intervals for the mean are very similar to those for IQM. For this reason, we have included them in the Appendix (B.2). We have added a description of the additional confidence intervals in the main body of the paper: *"Confidence intervals in Appendix B.2 show that PolyGRAD significantly outperforms model-free algorithms in terms of mean performance in addition to interquartile mean."*
> - We have added additional concurrent work to the related work.
>
> We did not train the policy on the real trajectories for the following reasons. One reason is that we used on-policy reinforcement learning to train the policy in the world model. Therefore, we had to use on-policy trajectories for RL training, and the real dataset is off-policy (it was collected using previous policies).
>
> We decided to use on-policy reinforcement learning because a world model should be able to simulate imagined trajectories under any desired action distribution. The action distribution we usually wish to simulate is that of the current policy (i.e. we wish to generate on-policy trajectories). Therefore, showing our world model is compatible with on-policy RL verifies that it successfully generates on-policy trajectories. Furthermore, on-policy RL algorithms are generally stabler and simpler than off-policy algorithms. Model-based RL with off-policy RL algorithms that use a mixture of real and synthetic data has been demonstrated in several previous works, particular those in offline RL (e.g. MBPO, MOPO, MOReL, RAMBO).
>
> We also think it is preferable to train policies using only synthetic data due to how world models might be used in future work. There is a trend towards training large and general models on internet-scale datasets (i.e. "foundation" models). A similar trend is occurring in the context of world models. Once a very general world model has been trained, it would be desirable to reuse the same world model every time we wish the agent to plan or optimise for a new task. For very large datasets, it might be impractical for the agent to access the entire dataset every time it optimises for a new behaviour. Training only on synthetic data generated by the world model that is relevant to the task of interest mitigates this problem. The world model then acts as a compressed representation of the knowledge that the agent has about the world in the same way that an LLM compresses knowledge represented as text on the internet (without having to access the dataset at inference time).
>
> Kind regards,
>
> The authors